# Far away or yesterday? Shifting perceptions of time for political ends

**Andrew J. Dawson** *, **Scott A. Leith, Cindy L. P. Ward, Sarah Williams, Anne E. Wilson**

Department of Psychology, Wilfrid Laurier University, Waterloo, Ontario, Canada

* daws6340@mylaurier.ca

## Abstract

Voters evaluate political candidates not only based on their recent record but their history, often faced with weighing the relevance of long-past misdeeds in current appraisal. How should a distant transgression be taken to reflect on the present? Across multiple years, political figures and incidents, we found that people's subjective perceptions of time concerning political candidate's histories can differ radically, regardless of objective fact; political bias shapes people's perception of the time of things past. Results showed that despite equidistant calendar time, people subjectively view a favored politician's successes and opposing politician's failures as much closer in time, while a favored politician's failures and opponent's success seem much further away. Studies 1–3 tested the proposed phenomena across distinct (real and hypothetical) political contexts, while Study 4 tested the causal effects of temporal distance framing. Study 5 provided a final preregistered test of the findings. Overall, we demonstrate that partisans can protect their candidates and attack opponents by shifting their perception of time.

## Introduction

> "*That was the old me—abrasive and confrontational . . . I am no longer the person I once was*"
>
> • *U.S. Congressman Newt Gingrich, 1985*

> "*I believe that I am a much more disciplined, much more mature person than I was 12 years ago. . .*"
>
> • *Presidential hopeful Newt Gingrich, 2012*

Elections are often won or lost not only on a candidate's current credentials, strengths, and weaknesses, but also on their past glories and misdeeds, successes and failures. When election season rolls around, inevitably the checkered pasts of those contending for public office are put under the microscope for all to consider. Many candidates may worry about which long-past transgressions will see the light of day—from the drugs, affairs, and secretly fathered children; to the ill-considered tweets, now-unpopular opinions, or videotaped "locker-room talk" of a long-past era. In turn, voters are faced with the task of weighing this temporally distributed

**Data Availability Statement:** Datasets, syntax for analyses, survey materials, and the Study 5 preregistration can be found in the OSF repository (https://osf.io/3dvyc).

**Funding:** A.W. received two awards from the Social Sciences and Humanities Research Council of Canada (Grant #435-2013-1930 and #435-2019-1034; https://www.sshrc-crsh.gc.ca/home-accueil-eng.aspx) as well as an award from the Department of National Defence Research Initiative (#877-2019-0007; https://www.sshrc-crsh.gc.ca/funding-financement/programs-programmes/dnd-eng.aspx). A.D. received the Canada Graduate Scholarships – Doctoral (CGS D) from the Social Sciences and Humanities Research Council of Canada (#767-2021-2281; https://www.sshrc-crsh.gc.ca/home-accueil-eng.aspx). No sources of funding played any role in the study design, data collection and analysis, decision to publish, or preparation of the manuscript.

**Competing interests:** The authors have declared that no competing interests exist.

information to evaluate a candidate's fitness for public office–do long-past actions truly matter, and at what point are actions remote enough that they can be safely disregarded?

Previous work has examined the timing of real political scandals and has experimentally examined discounting of hypothetical scandals from the recent or distant past [1–3]. Although there is some evidence that the impact of scandals can fade with temporal distance, other factors seem to matter more than chronological time including how recently the scandal came to light, recent media attention, and even the type of infraction [1–3]. Past research does not offer a clear answer to whether or when the passage of time will heal political wounds. One reason results may be mixed is because although the actual passage of time may be relevant, people's psychological experience of time often diverges considerably from chronological time. The subjective perception of time–whether past political misdeeds seem close or far away–may play an important role in the degree to which partisans hold onto those harms or disregard them in the present. We propose that people's political affiliations might shape the way people subjectively position the timing of past events in order to highlight or de-emphasize their relevance to a candidate's current character. The judged relevance of a person's past actions is key to judgments of current character and predictions about future success. If past actions are viewed as relevant in the present, they should affect people's judgments of the candidate's current moral character; if wrongdoings are seen as no longer relevant, they can be forgotten.

Central to these notions of relevance is the subjective perception of time: when an event actually occurred in time may have little to do with how close or distant it seems [4]. Mounting evidence suggests that people's subjective sense of time is malleable–past events can seem like ancient history or like yesterday independent of their chronological timing. Phenomenological features of a memory (such as vividness, detail, accessibility) can affect its perceived distance as can motivational factors (people tend to feel closer to desirable than undesirable past events; see Wilson et al. [5]); in turn, perceptions of time can play a role in downstream judgments people make about how important past events are for evaluating a present state of affairs [6–8]. Events that feel as though they occurred long ago are generally seen as less relevant to the present, as though a statute of limitations on their continued impact has passed [8, 9].

Might people construe the history of favored and disliked political candidates in such a way that some past acts are selectively deemed relevant to current evaluations, while others are seen as ancient and irrelevant history? Social cognitive research has demonstrated that processing of information congruent with one's beliefs is facilitated, while incongruent information is inhibited [10]. Easily-processed information is likely to be more fluent (easily brought to mind, accessible); according to research on the accessibility bias [11], memories that are more easily brought to mind are judged to be more recent. This subjective time judgment parallels the phenomenology of chronological time- since memories decay over time, (objectively) older memories are more difficult to reconstruct. However, many other psychological processes can affect the accessibility of a past event, aside from the gradual erosion of memory over time. As a result, people's subjective sense of time may be orthogonal to chronological time–yet, if time is treated as a cue to event relevance, then a subjectively close or distant past event may be deemed relevant or irrelevant respectively, without regard to the passage of calendar time [8]. From these principles, it follows that past political information congruent with one's beliefs may be easier to process and thus will be judged as more recent in time, while incongruent information will be experienced as less fluent, and thus judged as more temporally distant.

Motivated reasoning may also contribute to the tendency to systematically reconfigure the subjective timing of past events. People often approach political information with a pre-existing belief they wish to see supported (e.g., that their candidate is largely blameless, while the opponent candidate is evil incarnate). When faced with ambiguous information that could

support or discredit their desired conclusion, people interpret the evidence in a way that supports their goal [10, 12–14]. Stretching or condensing time–a conveniently malleable construct—can be one way to underscore or trivialize the significance of past glories or misdeeds [4, 15]. If an individual wishes to support the belief that something is irrelevant, recruiting "evidence" that the event seems sufficiently distant in time to disregard would be one path to the desired end state. Put simply, in the mental scramble to defend one's political beliefs, subjective temporal distance (and its implications for current relevance) is a conveniently elastic set of judgments one might shift to support a desired conclusion.

These processes would lead those backing different candidates to view the histories of those candidates in very different ways, with different implications for their current trustworthiness, moral goodness, competence, and likeability. Overall, we predict that people will perceive a favored candidate's past failures and transgressions as more distant in time, while an opposing candidate's (equidistant) misdeeds will seem much closer in time. Conversely, a favored candidate's successes should seem recent, while an opposing candidate's successes should seem temporally remote. In turn, more subjectively recent events are predicted to seem more important and relevant, while more subjectively distal events seem less relevant.

We seek to test the existence of this proposed phenomenon and its generality across elections and contexts, and to examine the hypothesized process by which subjective proximity of distance informs perceived relevance. This extends previous research demonstrating that subjective time matters for personal and social identity and close relationships [6, 7, 16] by demonstrating how shifts in subjective time can be a mechanism by which to attack or defend a person or group other than oneself, such as a politician.

## Precursor studies

In an effort to be maximally transparent, three studies from a prior paper investigating implicit theories of change [17] are included in the supporting information to this paper. These studies focused on the 2011 Canadian Federal Election, the Obama Administration in 2011, and the 2015 Canadian Federal Election respectively. Although subjective time was not the main variable of interest when the studies were conducted, item(s) measuring subjective temporal distance were included after the primary study materials, on a separate page (either paper or web), as an exploratory variable and not reported in the original paper. Exploratory analyses of those studies inspired the current set of studies explicitly designed to test this phenomenon. Given that these three studies allow us to test the replicability of the hypotheses and extend its generalizability across time and context, we chose to report these results in the supporting information.

Within these three precursor studies, we consistently found that when voters read negative information about a candidate's past, opponents of that candidate felt that this information was subjectively closer than supporters, who felt it was farther away. We also found that subjective time went on to predict judgments of current relevance of the past wrongdoings. These three precursor studies provide initial evidence that people may shift their view of time in ways that both attack opponents' current character and shield their favored candidates from harsh judgment. Over our five main studies, we conceptually replicate these initial findings using data newly collected to test these hypotheses a priori. This set of studies was conducted over multiple years, mostly prior to the emergence of strong preregistration norms. For this reason, we only preregistered our final and most recent study.

## Study 1

In Study 1, we examine the U.S. two weeks before the 2016 General Election vote. The 2016 U.S. election saw a highly divided country, in which many questionable events in the decades-old

past histories of both Hillary Clinton and Donald Trump–from Trump's genital grabbing to Clinton's "superpredator" rhetoric—were covered extensively in the media. This election offered an opportunity to examine the role of subjective time in a context where the relevance of the past was clearly a matter of contention; thus, we drew our study materials from real events. We were primarily interested in candidates' past failings being weighed in the present and sought to increase generalizability by selecting three past negative events for each candidate, with the intention to collapse across events if no meaningful differences emerged. We tried to select events that both opponents and supporters would acknowledge as factual and unflattering. We also sought to include a positive valence condition for comparison, but had more difficulty finding good examples of positively valenced events from both candidates' past (especially ones that both supporters and opponents would accept as factual and positive). We chose to select one positive event for each candidate from a roughly comparable past point in time. We recognize the imbalance in study design but chose to prioritize having a good representation of negative incidents committed by both candidates. We predicted that we would replicate the exploratory findings of our precursor studies, that people will shift subjective time to draw opponent failings close and push opponent successes away; and conversely push the shortcomings of a favored candidate into the distant past and draw their past glories nearer to the present.

## Method

**Participants and power.**   We recruited 686 American residents from Amazon's Mechanical Turk to participate in our on-line study in exchange for $0.75 dollars. Ethics approval was obtained from the Wilfrid Laurier University Research Ethics Board (Reference #4729). Participants provided written consent. As this study examines the effects of party loyalty and political bias, the Independent and undisclosed participants (n = 117) were excluded from analyses. Forty-eight participants were also excluded for failing the manipulation check (the subject of the article participants read), and 36 participants were excluded for failing at least one of two attention checks instructing participants to select a specific number. The final sample contained 492 U.S. citizens (314 Democrat, 178 Republican; 253 females, 237 males and 2 undisclosed; $M_{age}$ = 38.24, $SD$ = 12.32, Range 19–79). We did not conduct a priori power analyses (though we aimed for reasonably large sample sizes whenever possible). Instead we report the sensitivity analyses. According to G*Power 3.1 [18] this sample size was sufficiently sensitive to detect an interaction with an effect size of $f$ = 0.13 ($\eta_p^2$ = 0.02) with 80% power and $\alpha$ = 0.05.

**Procedure.**   This study was completed online in one session. Participants first completed demographics information, which included questions regarding age, gender, political orientation, and who they planned to vote for in the 2016 US election. Participants were randomly assigned to read either a *negative* or a *positive* past action by either Hillary Clinton (the Democratic candidate in the U.S.) or by Donald Trump (the Republican candidate in the U.S.). There were three negative incidents and one positive incident for each candidate for a total of eight events. All incidents were real, but the descriptions were written for the study. After collapsing across the negative action conditions, the experimental design was a 2 (valence: negative vs. positive) x 2 (preferred candidate: Clinton vs. Trump) x 2 (candidate: Clinton vs. Trump) between-subjects design. We used real past incidents for both candidates, thus by necessity they differed qualitatively and in timing. Clinton was criticized starting in 2014 for using an email account housed on a private server, beginning in 2009; for making a statement labeling some young, mainly black youths as "superpredators" in 1994; and for supporting the Iraq War in 2002. Trump was criticized for a recording of him boasting about groping women without their consent in 2005; for saying that Senator John McCain wasn't a war hero, stating

"I like people that weren't captured", in 2015; and for making statements indicating that he benefitted financially following a 2009 bankruptcy that left others in financial distress. Clinton was praised for her actions between 2001 and 2004 that helped 9/11 first responders; Trump was praised for his actions between 2002 and 2005 that caused his show The Apprentice to become a success. The real timeframe for each incident was clearly indicated in each of the respective descriptions; see S5 Table in S1 File for events and descriptions. After reading their randomly assigned incident participants indicated how far away in time the incident felt to them using a slider bar measure (1 = *Feels very recent*; 100 = *Feels very long ago*). The subjective time measure originally included a second item (1 = *Feels like yesterday*; 100 = *Feels like ancient history)* but we decided to exclude it because, as a reviewer pointed out, "ancient history" is a loaded term that is often used to dismiss past events. The "ancient history" item always came after the more neutrally worded item. See supporting information for results with the two-item measure, which did not differ meaningfully from the analyses reported here. Next, participants reported the current relevance of the candidate's past actions on four items (α = .93). The questions (rated from 1 = *Not At All* to 7 = *Very Much*) were "Do you believe that this candidate's past actions should affect their current standing in the public eye?" "How relevant is this candidate's past actions to your current opinion of them as a politician?" "How much do you think that this candidate's past actions reflect their current character?" "This past incident has no bearing on who this person is today;" [reverse scored item]. Finally, as a check of our valence manipulation, we then included two items evaluating the past act/event (1 = *Good*; 7 = *Bad* and 1 = *Moral*; 7 = *Immoral*), $r_{SB}$ = .97. (We measure the reliability using the Spearman-Brown coefficient, which is recommended for two-item scales [19]).

## Results and discussion

**Manipulation check.** Participants who read the negative incidents rated the incidents as significantly worse overall than those who read the positive incidents $F(1, 483)$ = 357.51, $p <$ .001, $\eta_p^2$ = .43. See supporting information for additional details. Estimated marginal means and standard errors for this analysis can be found in Table 1.

**Subjective temporal distance.** We hypothesized that supporters would distance the negative incidents while opponents would distance the positive incidents. We also predicted that supporters would distance negative incidents relative to positive incidents, and predicted the opposite effect for opponents. We conducted a 2 (valence: negative vs. positive) x 2 (political party: Democrat vs. Republican) x 2 (candidate: Clinton vs. Trump) ANOVA with subjective temporal distance as the DV controlling for the dates of the actual events. The results are presented in Fig 1. See supporting information for details on the different negative events examined separately.

**Table 1. Estimated marginal means and standard errors by cell for perceived valence of past event.**

| | Positive Incident | | | Negative Incident | | | Total | | |
|---|---|---|---|---|---|---|---|---|---|
| | Trump Incident | Clinton Incident | Total | Trump Incident | Clinton Incident | Total | Trump Incident | Clinton Incident | Total |
| **Clinton Voters** | 4.37 (0.20) | 6.52 (0.19) | 5.44 (0.14) | 1.75 (0.14) | 3.74 (0.14) | 2.74 (0.10) | 3.06 (0.12) | 5.13 (0.12) | 4.10 (0.08) |
| **Trump Voters** | 6.10 (0.26) | 5.31 (0.28) | 5.70 (0.19) | 3.10 (0.19) | 2.97 (0.18) | 3.04 (0.13) | 4.60 (0.16) | 4.14 (0.17) | 4.37 (0.11) |
| **Total** | 5.23 (0.16) | 5.91 (0.17) | 5.57 (0.12) | 2.42 (0.12) | 3.35 (0.11) | 2.89 (0.08) | 3.83 (0.10) | 4.63 (0.10) | 4.23 (0.07) |

Numbers in parentheses correspond to standard errors. Higher scores mean a more positive evaluation of the event.

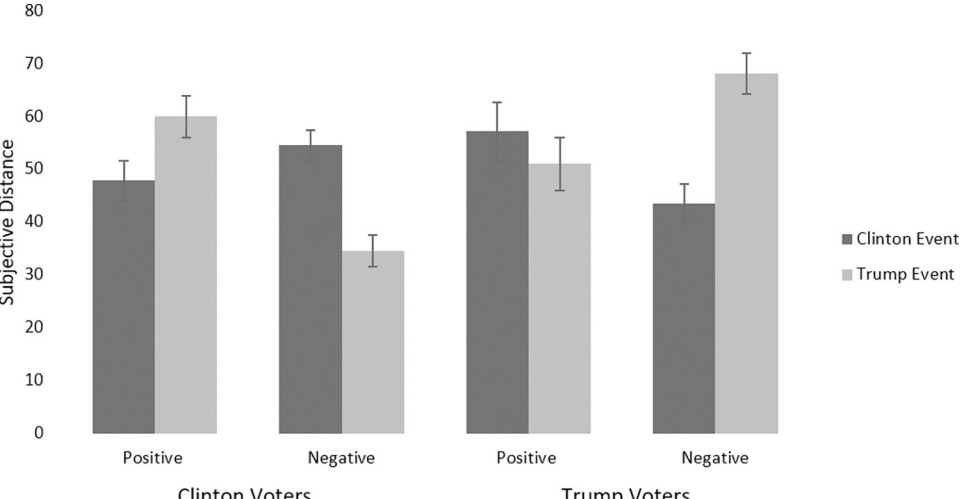

**Fig 1. Relationship between voter affiliation, candidate, and event valence on subjective temporal distance.** Bars represent standard errors.

As predicted, this revealed a significant 3-way interaction; $F(1, 483) = 31.62$, $p < .001$, $\eta_p^2 = .06$.

Clinton supporters rated Clinton's negative incidents as significantly more distant than Trump's negative incidents, $p < .001$, whereas Clinton's positive incident was rated significantly closer than Trump's positive incident, $p = .025$. While Clinton's negative incidents were not rated as any different than her positive incident, $p = .143$, Trump's negative incidents were rated as significantly closer in time than his positive incident, $p < .001$.

Similar patterns emerged for Trump supporters, who rated Trump's negative incidents as significantly more distant than Clinton's negative incidents, p $< .001$, and significantly more distant than Trump's positive incident, $p = .008$, while Clinton's negative incidents were seen as significantly closer in time than her positive incident, $p = .037$.

Overall, results support the hypotheses that, regardless of calendar time, partisans distance the failings of their favored candidates while viewing opponents' missteps as close in time, whereas their favored candidates' successes were viewed as more proximal than those of opponents.

**Current relevance.** As with Precursor Studies 1–3, we examined the degree to which supporters vs. opponents rated how relevant the candidates' past actions were to the present. A 2 (valence: negative vs. positive) x 2 (political party: Democrat vs. Republican) x 2 (candidate: Clinton vs. Trump) ANOVA with current relevance as the DV while controlling for the dates of the actual events. which revealed the predicted 3-way interaction; $F(1, 483) = 108.47$, $p < .001$, $\eta_p^2 = .18$. The results are presented in Fig 2.

As expected, Clinton/Democrat supporters indicated that Trump's past negative incidents reflect on who he is today significantly more than his past positive incident, $p < .001$, and significantly more so than Clinton's past negative incidents should reflect on her, $p < .001$. in contrast, Clinton's past positive incident was rated significantly more relevant to the present by her supporters than her past negative incidents; $p < .001$, and significantly more relevant than Trump's past positive incident; $p = .008$.

The same pattern emerged for Trump/Republican supporters who indicated that Clinton's past negative incidents should reflect on who she is today significantly more than her past positive incident, $p < .001$, and significantly more so than Trump's past negative incidents should

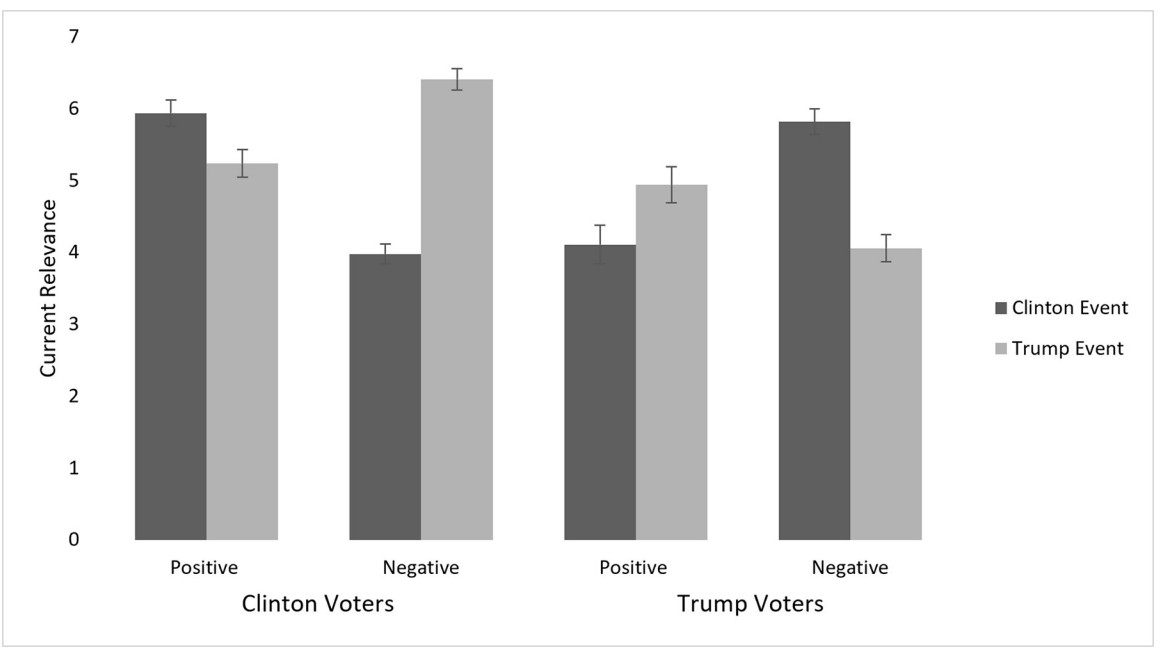

**Fig 2. Relationship between voter affiliation, candidate, and event valence on current relevance.** Bars represent standard errors.

reflect on him, $p < .001$. Whereas Trump's past positive incident was rated significantly more relevant to the present by his supporters than his past negative incidents; $p = .005$, and significantly more relevant than Clinton's past positive incident; $p = .024$.

## Study 2

In Study 2, we revisited the U.S. one year after the 2016 General Election. Given the contentious nature of the election, and the ensuing year, we expected that overall the election of Donald Trump would be seen as a positive event for his supporters, and a negative event for Hillary Clinton supporters. However, we also sought to shift valence experimentally. We predicted that the narrative surrounding the election could be framed in a manner that would be more appealing (and therefore positive) to Trump supporters–that of a glorious, anti-establishment victory–or one that appeals to Clinton supporters–highlighting the doubts and concerns surrounding the legitimacy of the election. We predicted that Trump supporters would feel particularly close to the election when it was portrayed as a moment of glory, whereas we expected that Clinton voters would feel relatively closer to the election when it was portrayed as more tenuous and less legitimate.

## Method

**Participants and power.** We recruited 458 American residents from Amazon's Mechanical Turk, in exchange for $1.00. Ethics approval was obtained from the Wilfrid Laurier University Research Ethics Board (Reference #4729). Participants provided written consent. Participants were excluded for not having completed the study materials (n = 6); attrition rates were equivalent across conditions and voters. Participants who did not vote for, or express a desire to have voted for, Trump or Clinton were excluded, including those that identified another preferred candidate, as well as those who did not specify a candidate (n = 75). We also excluded all participants who reported feeling "very dissatisfied" with their vote at present, as

those participants could no longer be classified as supporters of the candidate they voted for (n = 19). Finally, participants were also excluded for failing at least one of two attention checks instructing participants to select a specific number (n = 38). The final sample contained 327 U. S. citizens (197 Clinton supporters, 130 Trump supporters; 187 females, 136 males and 4 undisclosed; $M_{age}$ = 37.79, SD = 12.67, Range 20–75). According to G*Power 3.1 [18] this sample size was sufficiently sensitive to detect an effect size of $f$ = 0.17 ($\eta_p^2$ = 0.03) with 80% power and $\alpha$ = 0.05.

**Procedure.** This study was completed online in one session. Participants completed demographics information, including age, gender, race, and political ideology and voting partisanship. Participants were then randomly assigned to one of three conditions. In the *positive* election condition, participants read a passage created for the study where Donald Trump's election was described as a turning point, allowing Americans' displeasure with "business as usual" to finally be heard. In the *negative* election condition, the passage questions the decisiveness of the election, describing Trump's loss of the popular vote, and concerns about Russian interference in the election. Finally, in the *control* condition, participants only received the information that Donald Trump won the November 8th election, with no additional information. See S7 Table in S1 File for the complete text of passages.

After reading the passage about the 2016 elections, participants indicated how close or far the election felt, using the same measure as Study 1 (1 = *Feels very recent*; 100 = *Feels very long ago*). Again, because of reviewer concerns about the loaded nature of a second item (with the endpoint "ancient history"), results of a two-item version of this measure are reported in the supporting information rather than the main manuscript. Patterns of results are very similar conducted both ways. In addition to the key measures for hypothesis tests, we measured two other aspects of subjective time for exploratory purposes: the felt *duration* of the Trump presidency to date, and the subjective distance to the next (*future*) election. We had no specific hypotheses related to these variables, which are conceptually distinct from subjective distance to the (past) election. Because the manipulation of valence would only be expected to alter people's view of the election itself (how flattering or disparaging it was to Trump) but not the ongoing experience of Trump's presidency, we had no reason to think the manipulation of past election valence would affect these other measures of subjective time. To capture subjective duration, we asked participants how long it felt like Trump had been president (1 = *Not long at all*; 100 = *A very long time*) and to assess time to the next election we asked how close or far the *next* American presidential election, in 2020, seemed to them (1 = *Feels very near*; 100 = *Feels very distant*). Although we had no hypotheses for these measures, past research demonstrates that "time flies when you're having fun" while unpleasant events tend to "drag on" subjectively [20]. Since Clinton voters would likely experience the Trump presidency as more unpleasant, is it possible that they would perceive the subjective duration as longer than Trump voters.

## Results and discussion

**Subjective temporal distance from the 2016 election.** We again hypothesized that opponents of the candidates would distance positive events relative to supporters, while the reverse would occur for negative events. However, because an election victory is a positive event for supporters, but a negative event for opponents, we predicted that a *positive* election passage would be a positive event for supporters but a negative one for opponents, and that the opposite would be true of the negatively-valenced election passage. Thus, we hypothesized that Clinton supporters would perceive the positively-valenced election passage as a negative event, and distance it accordingly, relative to Trump supporters, who would perceive the election

**Table 2. Estimated marginal means and standard errors by cell for measures of subjective temporal distance.**

| | Positive Framing | Control | Negative Framing | Total |
|---|---|---|---|---|
| **Subjective Temporal Distance from 2016 Election** | | | | |
| **Clinton Voters** | 47.85 (3.66) | 38.24 (3.80) | 32.89 (3.98) | 39.66 (2.20) |
| **Trump Voters** | 35.27 (5.07) | 39.50 (4.45) | 45.96 (4.60) | 40.24 (2.72) |
| **Total** | 41.56 (3.13) | 38.87 (2.93) | 39.43 (3.04) | 39.95 (1.75) |
| **Subjective Temporal Length of Trump Presidency** | | | | |
| **Clinton Voters** | 57.13 (3.54) | 52.94 (3.67) | 57.93 (3.85) | 56.00 (2.13) |
| **Trump Voters** | 33.14 (4.90) | 34.27 (4.31) | 38.93 (4.45) | 35.45 (2.63) |
| **Total** | 45.13 (3.02) | 43.61 (2.83) | 48.43 (2.94) | 45.72 (1.69) |
| **Subjective Temporal Distance from 2020 Election** | | | | |
| **Clinton Voters** | 81.27 (2.79) | 80.77 (2.89) | 82.77 (3.03) | 81.60 (1.68) |
| **Trump Voters** | 67.73 (3.86) | 67.71 (3.39) | 73.96 (3.50) | 69.80 (2.07) |
| **Total** | 74.50 (2.38) | 74.24 (2.23) | 78.36 (2.32) | 75.70 (1.33) |

Numbers in parentheses correspond to standard errors.

more positively. When Trump's victory was portrayed as more questionable, and thus in a more negative light, however, we expected this pattern to reverse.

We conducted a 3 (valence: positive vs. negative vs. control) x 2 (voting preference: Clinton vs. Trump) ANOVA with subjective temporal distance from Election Day 2016 as the DV. Estimated marginal means and standard errors for this analysis can be found in Table 2. As predicted, we found a significant 2-way interaction between valence and voting preference, $F (2, 321) = 4.32$, $p = .014$, $\eta_p^2 = .03$; there was no main effect of valence or of voting preference.

Within the negatively-valenced election description condition, Clinton supporters indicated that the original election felt significantly closer to present than did Trump supporters, p = .033, while in the positively-valenced condition, Trump supporters rated the election as feeling closer to present than did Clinton supporters, p = .045. There was no significant difference in the control condition, $p = .830$.

Differences were primarily found among Clinton supporters, who indicated that the positively-valenced election passage felt significantly more distant than the negative passage, $p = .006$ and marginally further than in the control condition, $p = .070$. The negative and control conditions did not differ, $p = .332$. For Trump supporters, there was no difference between the positive and negative passage, $p = .119$, neither of which differed from control $p = .531$, 313.

**Other measures of subjective time.** Our primary hypothesis was that Clinton and Trump voters would show different patterns of subjective distance to the past election when it was framed in a way that was flattering or unflattering to the political candidate; results supported this prediction. We also included other exploratory measures of subjective time: perceived subjective duration and distance to the next (future) election. Estimated marginal means and standard errors for these analyses can be found in Table 2.

Because our predicted interaction for subjective distance to the 2016 election was contingent on the manipulation of valence for a specific event (framing the election result in flattering or questionable terms), we did not predict the same pattern for these distinct aspects of subjective time. Framing the 2016 election in different ways would be unlikely to affect people's experience of the ongoing presidency or expectations about the future election. Accordingly, no interaction was observed for the measures of the subjective duration of the Trump

presidency, $F(2, 321) = 0.25$, $p = .778$, $\eta_p^2 = .002$, nor the subjective distance to the next election, $F(2, 321) = 0.32$, $p = .730$, $\eta_p^2 = .002$. For both measures, only voter identification was a significant predictor of subjective *duration* of the presidency, $F(1, 321) = 36.84$, $p < .001$, $\eta_p^2 = .10$, and subjective distance to the next election, $F(1, 321) = 19.58$, $p < .001$, $\eta_p^2 = .06$. Valence condition was not significant for duration, $F(2, 321) = 0.72$, $p = .486$, or future election distance $F(2, 321) = 1.00$, $p = .369$, $\eta_p^2 = .006$. Overall, Clinton supporters felt that Trump had been president for a longer subjective duration and viewed the next election as subjectively further away. Such findings may be seen as consistent with past research showing that unpleasant events drag on while "time flies" subjectively for pleasant durations [20]. Since Clinton voters are likely to experience the Trump presidency as more unpleasant than Trump voters, this may account for why the duration was perceived to be longer, and could perhaps explain why it felt like a longer wait until the next election. However since past research on subjective duration has not examined whether may also extend in the future direction, this interpretation is especially speculative.

## Study 3

So far we have shown that voters will alter the subjective temporal distance of a politician's past actions according to the actions' valence, and the politician's party affiliation, and that differences in subjective temporal distance predict differences in beliefs about the current applicability of past information. However, we have drawn on real-world figures and events. While this lends realism to our research, it can introduce many outside factors beyond our control that could affect results. Thus, in Study 3, we introduced Congressman Paul Bosch–a fictitious U.S. politician. By using Mr. Bosch as our political figure of interest, we were able to keep the politician constant while only changing his political affiliation. We note that as Mr. Bosch was explicitly fictitious, we expected our effects to decrease somewhat in magnitude.

### Method

**Participants and power.**   Five hundred and twenty-three American residents recruited from Amazon's Mechanical Turk participated in our on-line study in exchange for $0.75 dollars. Ethics approval was obtained from the Wilfrid Laurier University Research Ethics Board (Reference #3953). Participants provided written consent. At the study's outset, we asked participants "If an election were held tomorrow, which party would you vote for?" (Democrat, Republican, Other, I wouldn't vote; participants selecting 'Other' were invited to specify their party). Inclusion criteria was determined on the basis of whether or not participants gave clear enough partisanship information that they could be classified as either a "supporter" or "opponent." For instance, when Bosch was described as a Democrats, participants who selected "Democrat" were included as supporters, and those who selected "Republican" or "Other" and stated a coherent preference were labeled opponents. Participants who explicitly indicated they do not vote (n = 46) and participants who selected 'Other' but gave no or a vague indication of their political affiliation (e.g., "It depends"; n = 47) could not be classified as a supporter or opponent of the Democrat or Republican target candidate (Bosch); however, for respondents randomly assigned to a neutral (no party specified) Bosch, no participants were excluded on the basis of partisanship because they did not have to be categorized as supporters or opponents. In addition, 8 inattentive responders (i.e., clicking '6' for every question) were excluded. The final sample consisted of 454 U.S. residents (313 Democrats, 110 Republicans, 31 Other; 279 males, 171 females, 4 undeclared; $M_{age} = 33.86$, $SD = 11.41$, Range 18–75). According to G*Power 3.1 [18] this sample size was sufficiently sensitive to detect an effect size of $f = 0.15$ ($\eta_p^2 = 0.02$) with 80% power and $\alpha = 0.05$.

**Procedure.** This study was completed online. Participants first completed demographics information as in previous studies.

Participants were informed that they were about to read a fictitious scenario about the upcoming presidential primaries, and were told to report how they would react if this were to happen. The manipulation had a 3 (Bosch affiliation: Democrat vs. Republican vs. Not Mentioned) × 2 (review valence: positive vs. negative) design; thus, participants were randomly assigned to read one of six possible versions of the following vignette:

*Rep. Paul Bosch (D) [(R), no mention], congressman for the 18th district in Florida, has won the Democratic [Republican, no mention] primary and will run for President of the United States in 2016. His rapid rise in popularity from near-obscurity, and his decisive victory in the primaries has been attributed to his passion, charisma and his popularity with grassroots Democrat [Republican, no mention] voters.*

*In the wake of his victory, reporters and watchdog groups have been busy investigating Bosch's past. One major finding has surfaced: The independent House of Representatives Performance Review Committee, with appointees from across the political spectrum from right to left, conducted a complete review of each representative's performance in 2008. Although it received little media attention at the time, they have re-released their damning [glowing] short report on the performance of Paul Bosch.*

*The bi-partisan committee members noted that "Mr. Bosch's performance during the turning point of 2008 was marked by an inability to get momentum [a great momentum] on the issues he championed during election time," and that ". . .his statements were inconsistent [very consistent] with the values he campaigned on–he changed his priorities more than once, leading to confusion about what exactly he stood for [he set clear priorities, and there was no confusion about what he stood for]."*

Overall, Mr. Bosch received either an 'A' (positive condition) or a 'C' (negative condition) grade for his performance in 2008, a similar method to that used in the precursor studies.

After reading the performance review, participants rated the reviews from 1 (*very negative*) to 7 (*very positive*). Following this, participants indicated how far away in time the year 2008 *felt* with a two-item measure (1 = *Feels Like Yesterday*; 10 = *Feels Like a Long Time Ago* and 1 = *Feels Very Close*; 10 = *Feels Very Distant*; $r_{SB}$ = .94). Participants indicated their agreement with four statements assessing the current relevance of the performance review, the first three adapted from the previous studies and a fourth that asked "How do you think the 2008 performance review reflects on Mr. Bosch in terms of who he is now?" (α = .92). Additional outcome measures included a person perception measure assessing participants' opinions of Bosch's likeability, leadership, and trustworthiness on 7-point scales (α = .93) and a single item assessing their inclinations to vote for him in a hypothetical scenario (1 = *Not Likely At All*; 7 = Extremely Likely). See Table 3 for inferential statistics on the additional measures and Table 4 for estimated marginal means and standard errors.

## Results and discussion

As in previous studies we examined voters who could clearly be considered motivated *supporters* or *opponents* of the featured candidates; voters were grouped into three categories for the remainder of the analyses: *supporter* (i.e., Democrats rating a Democrat Bosch), *opponent* (e.g., Democrats or "Other" voters rating a Republican Bosch), and *neutral* (voters of any affiliation

**Table 3. Main effects and interactions for additional outcome measures.**

| Factor | $F, p, \eta_p^2$ |
| --- | --- |
| **Person Perception** | |
| Review Valence | $F(1, 448) = 220.95, p < .001, \eta_p^2 = .33$ |
| Voter Group | $F(2, 448) = 40.17, p < .001, \eta_p^2 = .15$ |
| Review Valence × Voter Group | $F(2, 448) = 0.41, p = .662, \eta_p^2 = .002$ |
| **Voting Intentions** | |
| Review Valence | $F(1, 445) = 125.32, p < .001, \eta_p^2 = .22$ |
| Voter Group | $F(2, 445) = 99.82, p < .001, \eta_p^2 = .31$ |
| Review Valence × Voter Group | $F(2, 445) = 0.90, p = .406, \eta_p^2 = .004$ |

rating the neutral Bosch). A 2 (review valence: negative vs. positive) x 3 (voter: supporter vs. opponent vs. neutral) ANOVA was run for each of the main dependent variables.

**Manipulation check.** Those reading a negative review perceived it as more negative than those reading a positive review, $F(1,448) = 1505.93, p < .001, \eta_p^2 = .77$. See supporting information for additional details. Estimated marginal means and standard errors for this analysis can be found in Table 4.

## Subjective temporal distance

We that supporters would distance a negative review relative to controls, while opponents would distance the positive review relative to controls. We also predicted that supporters would distance negative reviews relative to positive reviews, and predicted the opposite effect for opponents. There was a significant valence by voter interaction; $F(2,448) = 4.44, p = .012, \eta_p^2 = .02$. Within the negative condition, opponents indicated that the review seemed significantly closer in time than supporters, $p < .001$, and controls, $p = .040$. Supporters rated the negative review as marginally more distant than controls, $p = .060$. There were no significant differences between voter groups within the positive condition (all $ps > .05$). Estimated marginal means and standard errors for this analysis can be found in Table 4.

**Current relevance.** We ran a 2 (valence: negative vs. positive) x 2 (voter: supporter vs. opponent vs control) ANOVA with the combined relevance measure, which revealed the expected interaction, $F(2,448) = 13.47, p < .001, \eta_p^2 = .06$. Within the negative condition, opponents indicated that the review was significantly more relevant than supporters, $p < .001$, but did not differ from controls, $p = .325$. Supporters rated the negative review as less relevant than controls, $p = .006$. Within the positive condition, opponents rated the review as significantly less relevant than supporters, $p < .001$, and no different from controls, $p = .188$. Supporters rated the positive review as significantly more relevant than controls, $p = .026$.

**Table 4. Estimated marginal means and standard errors by cell for each of the dependent variables.**

| | Supporter | | | Neutral | | | Opponent | | | Total | |
| --- | --- | --- | --- | --- | --- | --- | --- | --- | --- | --- | --- |
| | Positive | Negative | Total | Positive | Negative | Total | Positive | Negative | Total | Positive | Negative |
| Perceived Valence | 7.60 (0.12) | 3.75 (0.12) | 5.68 (0.08) | 7.43 (0.11) | 3.80 (0.11) | 5.61 (0.08) | 7.12 (0.12) | 3.57 (0.11) | 5.34 (0.08) | 7.38 (0.07) | 3.71 (0.07) |
| Subjective Time | 6.40 (0.24) | 6.97 (0.24) | 6.69 (0.17) | 6.73 (0.23) | 6.34 (0.23) | 6.53 (0.17) | 6.49 (0.25) | 5.65 (0.23) | 6.07 (0.17) | 6.54 (0.14) | 6.32 (0.14) |
| Current Relevance | 5.27 (0.14) | 4.00 (0.14) | 4.67 (0.10) | 4.84 (0.14) | 4.62 (0.14) | 4.73 (0.10) | 4.58 (0.15) | 4.80 (0.14) | 4.69 (0.10) | 4.90 (0.08) | 4.50 (0.08) |
| Person Perception | 5.57 (0.12) | 3.98 (0.12) | 4.77 (0.09) | 4.92 (0.12) | 3.51 (0.12) | 4.21 (0.08) | 4.37 (0.13) | 2.99 (0.12) | 3.68 (0.09) | 4.95 (0.07) | 3.49 (0.07) |
| Voting Intentions | 5.51 (0.16) | 4.08 (0.16) | 4.80 (0.11) | 4.58 (0.15) | 2.92 (0.15) | 3.75 (0.11) | 3.16 (0.16) | 1.92 (0.15) | 2.54 (0.11) | 4.42 (0.09) | 2.98 (0.09) |

Numbers in parentheses correspond to standard errors.

Opponents did not rate the positive review as significantly different than the negative review, *p* = .259. Supporters rated the positive review as significantly more relevant than the negative review, *p* < .001. In the neutral control, there was no difference between condition, *p* = .238. Estimated marginal means and standard errors for this analysis can be found in Table 4.

## Study 4

So far we have shown that partisans perceive the subjective timing of equidistant past political actions and events depending on what side of the aisle they are viewing from. In general, opponents' scandals and failings appear closer and more relevant than favored candidates, and the reverse is observed for laudable past events. Although we suggest that people may adjust their perceptions of subjective distance to support their desired view of the relevance of past political events to the present, we have not tested the causal effect of subjective distance on relevance or other judgments of candidates in the present. Past research has shown that subjective time can be altered by the external framing of an event [6]. In Study 4, we attempt to manipulate the subjective distance of a political scandal to examine its causal effect on judgments. To avoid the complexities of real-world political scandals, we again visit the past of hypothetical candidate Paul Bosch, and reveal a moral misdeed–marital infidelity—that he committed several years prior.

## Method

**Participants and power.** We recruited 407 American residents recruited from Amazon's Mechanical Turk participated in our online study in exchange for $0.40 dollars. Ethics approval was obtained from the Wilfrid Laurier University Research Ethics Board (Reference #3953). Participants provided written consent. Because the study design required knowledge of the participants' political leanings, those who did not disclose partisanship were excluded (n = 6). Because we thought judgments of marital infidelity may vary by gender, we also limited analyses to participants identifying as men and women and included gender as a covariate (2 excluded). Finally, participants were excluded if they were missing responses to key DVs (n = 23). This left a final sample of 374 participants (258 Democrats, 64 Republicans, 52 Other; 227 males, 147 females; $M_{age}$ = 30.63 *SD* = 11.29, Range 19–82). According to G*Power 3.1 [18] this sample size was sufficiently sensitive to detect an effect size of $f$ = 0.18 ($\eta_p^2$ = 0.03) with 80% power and α = 0.05.

**Procedure.** This study was completed online. Participants first completed demographics information as in previous studies. The study, conducted in 2013, had a 3 (Bosch affiliation; Democrat vs. Republican vs. Not Mentioned) × 3 (temporal descriptor; Close vs. Far vs. Neutral) design. Participants were randomly assigned to read one version of the following vignette about the (explicitly) fictional candidate. In all cases the affair was described as occurring in 2008, which was roughly five years prior to the present (the study was conducted in 2013). As is often the case with political candidates of rising prominence, old skeletons will be pulled out of closets. Because dishonesty (attempting to conceal a past affair) could complicate participants' impressions, we made it clear that the affair was revealed and acknowledged at the time it occurred, but was back in the news because of Bosch's current presidential ambitions.

***[Democratic; Republican; No mention]** Senator for the State of Florida, Paul Bosch is expected to announce that he will run for President of the United States sometime in 2013. On the wake of this information, reporters and watchdog groups have been busy investigating Bosch's past.*

*One major finding has surfaced*: Mr. Bosch was reported as having an affair outside of his marriage in 2008.

**[Close condition: *Not so long ago, still within this electoral cycle;*], [Far condition: *Quite a number of years ago, some time before being elected Senator*], [Control condition: *No temporal framing*]**, *a 2008 investigative report in the Miami Herald reported that Paul Bosch had an affair outside of his 12 year marriage to Carolyn Bosch, with a senior administrator in his office. At the time, the affair did not receive much national publicity. Mr. Bosch reluctantly confirmed the allegations at that time. He and Carolyn are still married today.*

Participants indicated how far away in time Bosch's infidelity *felt* using a slider bar (a line ranging from 1 = *Feels very close* to 100 = *feels like a long time ago*). Following this, participants reported their beliefs concerning the current relevance of his infidelity through two items: "Do you believe that Mr. Bosch's 2008 affair should affect his current standing in the public eye?" (1 = *Not At All*; 7 = *Very Much*) and "How relevant is Mr. Bosch's affair to your current opinion of him?" (1 = *Not At All Relevant*; 7 = *Extremely Relevant*), α = .90. As in Study 3 we included additional outcome measures assessing perceptions of Bosch's likeability, leadership, and trustworthiness (α = .81), as well as likelihood of voting for him in future.

At the end of the study participants were asked to recall the year that the 2008 affair took place. Although we did not expect perfect accuracy we wanted to ensure that there were no condition or party differences. Analyses revealed no significant differences in memory for the timing of the affair across experimental conditions or voter groups (all $ps < .05$).

## Results and discussion

As in Study 3 we combined participant party affiliation with Bosch party affiliation, resulting in two conditions of "supporter" (e.g., Democrats rating a Democrat Bosch) and "opponent" (i.e., A Republican rating a Democrat Bosch). Participants reading the vignette in which Bosch's affiliation was not mentioned were labeled "neutral". Participants indicating an "Other" non-Democrat or Republican allegiance (e.g., Libertarian, 'Ron Paul') were also labeled "opponent" for both Democrat and Republican Bosch conditions.

**Subjective temporal distance.** A 3 (voter group; supporter vs. opponent vs. neutral) × 3 (time condition; close vs. distant vs. control) ANCOVA examined the effects of the manipulations of subjective temporal distance. We included gender as a covariate (as with subsequent analyses) due to the nature of Bosch's transgression in the vignette. Only the main effect of time manipulation was significant, $F(2, 364) = 6.07$, $p = .003$, $\eta_p^2 = .03$. Post-hoc analysis revealed that participants in the distant condition rated Bosch's misdeed as significantly further in the past than participants in the control condition, $p = .008$, and the close condition, $p = .002$. Temporal distance ratings did not differ between those in the close and control conditions, $p = .591$. Estimated marginal means and standard errors for this analysis can be found in Table 5. As for all dependent variables, the pattern of results did not change if gender was removed as a covariate.

## Current relevance

Perceptions of current relevance revealed a significant main effect of time condition, $F(2, 364) = 3.27$, $p = .039$, $\eta_p^2 = .02$. Participants in the close condition judged the affair as more relevant to the present than those in the distant condition, $p = .013$. The neutral condition did not differ significantly from the close condition, $p = .352$, or the distant condition, $p = .125$. There was no main effect of voter group, $F(2, 364) = 0.46$, $p = .635$, $\eta_p^2 = .002$.

**Table 5. Estimated marginal means and standard errors by cell for each of the dependent variables.**

| | Supporter | | | | Neutral | | | | Opponent | | | | Total | | |
|---|---|---|---|---|---|---|---|---|---|---|---|---|---|---|---|
| | Close | Control | Dist. | Total | Close | Control | Dist. | Total | Close | Control | Dist. | Total | Close | Control | Dist. |
| Subjective Time | 62.42 (4.01) | 56.75 (4.16) | 70.97 (3.68) | 63.38 (2.28) | 55.00 (4.46) | 59.00 (4.11) | 69.66 (3.80) | 61.22 (2.28) | 56.44 (4.59) | 63.78 (4.39) | 64.54 (3.51) | 61.59 (2.42) | 57.96 (2.52) | 59.84 (2.44) | 68.39 (2.11) |
| Current Relevance | 3.02 (0.26) | 3.33 (0.27) | 3.29 (0.24) | 3.21 (0.15) | 3.53 (0.29) | 3.38 (0.27) | 2.53 (0.25) | 3.15 (0.15) | 3.91 (0.30) | 3.11 (0.28) | 3.04 (0.23) | 3.35 (0.16) | 3.49 (0.16) | 3.28 (0.16) | 2.95 (0.14) |
| Person Perception | 3.82 (0.16) | 3.82 (0.17) | 4.08 (0.15) | 3.91 (0.10) | 3.50 (0.18) | 3.58 (0.17) | 3.98 (0.16) | 3.68 (0.10) | 2.93 (0.19) | 3.39 (0.18) | 3.54 (0.14) | 3.29 (0.10) | 3.42 (0.10) | 3.60 (0.10) | 3.87 (0.09) |
| Voting Intentions | 3.98 (0.19) | 3.92 (0.20) | 4.16 (0.17) | 4.02 (0.11) | 3.56 (0.21) | 3.52 (0.19) | 4.17 (0.18) | 3.75 (0.11) | 2.44 (0.22) | 3.00 (0.21) | 3.29 (0.16) | 2.91 (0.11) | 3.32 (0.12) | 3.48 (0.12) | 3.87 (0.10) |

Numbers in parentheses correspond to standard errors.

These effects were qualified by an interaction, $F(4, 364) = 2.50$, $p = .045$, $\eta_p^2 = .03$ revealing that the distance manipulation was effective for those judging a politically neutral candidate or a political opponent but not among political supporters. Within neutral voters, we find that the distant condition produced lower ratings of relevance relative to the close condition, $p = .009$, and control condition, $p = .020$, while the close and control conditions did not differ, $p = .691$. Participants rating an opposing candidate rated the infidelity as more relevant in the close condition relative to the distant condition, $p = .020$, and (marginally) to the control condition, $p = .054$, while the distant and control conditions did not differ, $p = .829$. In contrast, among those rating a favored candidate, the close condition did not differ from the distant condition, $p = .434$, or the control, $p = .395$, nor did the distant and control conditions differ from each other, $p = .907$. Estimated marginal means and standard errors for this analysis can be found in Table 5.

**Additional analyses.** We also examined whether people's present judgments of the candidate and willingness to vote for them would be affected by the subjective distance framing of the past scandal. Not surprisingly, political supporters rated candidates more favorably than opponents. More importantly, subjective temporal framing of the scandal also affected people's judgments, with respondents judging candidates more favorably and being more willing to vote for them when the 2008 affair was described as further away in time. Estimated marginal means and standard errors can be found in Table 5 and inferential statistics are in Table 6.

## Study 5

Studies 1–3 revealed that the good and bad past deeds of political actors can at times be seen as very recent and relevant to the present and at other times be relegated to the distant, and irrelevant, past, and that subjective time perceptions appear to be highly contingent on political

**Table 6. Main effects and interactions for additional outcome measures.**

| Factor | $F$, $p$, $\eta_p^2$ |
|---|---|
| **Person Perception** | |
| Time Manipulation | $F(2, 364) = 5.83$, $p = .003$, $\eta_p^2 = .03$ |
| Voter Group | $F(2, 364) = 10.51$, $p < .001$, $\eta_p^2 = .06$ |
| Review Valence × Voter Group | $F(2, 364) = 0.60$, $p = .665$, $\eta_p^2 = .01$ |
| **Voting Intentions** | |
| Time Manipulation | $F(2, 364) = 7.01$, $p = .001$, $\eta_p^2 = .04$ |
| Voter Group | $F(2, 364) = 26.89$, $p < .001$, $\eta_p^2 = .13$ |
| Review Valence × Voter Group | $F(2, 364) = 1.25$, $p = .288$, $\eta_p^2 = .01$ |

motivations to either excuse or condemn. Study 4 demonstrated the causal role of subjective time perceptions on downstream judgments. The final study had a couple of additional goals. First, because the earlier studies were conducted prior to preregistration being commonplace, we sought to preregister a conceptual replication of past studies with a more current (and arguably more momentous) political event (the January 6, 2021 storming of the Capitol). The preregistration form is available on Open Science Framework (https://osf.io/3dvyc/?view_only= fe41bd0f690c40c782950749830047e7). A secondary goal was to explore whether the motivation to distance could be altered by increasing or reducing group identity threat.

On January 6[th], 2021, a large group of protesters forced their way into the United States Capitol building to stop the session of congress that would ratify Joe Biden's victory in the 2020 US presidential election [21, 22]. The protesters were largely Republican voters and supporters of the incumbent Donald Trump, who had been falsely claiming his loss in the election was due to a rigged process [22–24]. Several Republican lawmakers aligned themselves with Trump's claims [25, 26]. Upon entering the building, the rioters assaulted Capitol police officers and attempted, unsuccessfully, to locate lawmakers who were being evacuated [27, 28]. Five people died during or shortly after the assault, and many more were injured [29, 30]. The relative roles of both mainstream Republicans and extremist fringe groups was a topic of interest (and contention) following the attack [31–33]. The purpose of our final study was to examine motivated shifts in the subjective time of this recent event of historical significance while adding preregistered analyses to the current package. The study was conducted less than five months after the January 6 incident.

We expected that the Capitol attack would elicit the same motivated shifts in subjective time observed in the past studies. Because the event primarily reflects negatively on Republican leaders and voters, we predicted that Republican participants would view the Capitol attack as farther away and less relevant in the present than Democrats. We also attempted to manipulate the focus of responsibility for the event by highlighting in two different versions of events either the role of Republicans or of fringe groups (such as white supremacists). Although both frames still reflect poorly on Republicans more than Democrats and do not alter the predicted party differences, we also predicted that Republican participants might be less motivated to distance and downplay the relevance when a fringe rather than their party was framed as the primary perpetrators.

## Method

**Participants and power.**   From May 26–28, 2021, participants from CloudResearch self-selected into the study and were compensated $2.00 USD for their participation. Ethics approval was obtained from the Wilfrid Laurier University Research Ethics Board (Reference #6840). Participants provided written consent. We aimed to recruit roughly equal numbers of Republicans and Democrats. Our initial sample included 855 participants. Four participants were removed because they retracted consent at the end of the study. Sixty-two participants failed at least one of two attention checks instructing participants to select a specific number, and were excluded. We also asked at the end of the survey if participants believed their answers were honest and accurate, and therefore should be included in our analyses. Three participants indicated that we shouldn't and were excluded. Finally, we only included participants who identified within the survey as Republicans or Democrats. We excluded 107 participants who identified a different voter affiliation. The final sample included 707 US citizens (376 Democrat, 331 Republican; 387 female, 314 male, 4 other; $M_{age}$ = 43.71, $SD$ = 14.02, Range 18–83). According to a G*Power sensitivity analysis [18], this sample size was sufficiently sensitive to detect an effect size of $f$ = 0.10 ($\eta_p^2$ = 0.01) with 80% power and $\alpha$ = 0.05.

**Procedure.** This study was completed online. Participants provided demographic information (age, gender, race, political orientation) before the manipulation.

Participants were told that they would be randomly assigned to read about an event that occurred in the United States, from a description taken largely verbatim from an information site similar to Wikipedia that aggregates sources on events in the United States. In fact, all participants read about the storming of the United States Capitol on January 6[th], 2021. Participants were further randomly assigned to one of two focus conditions. Both conditions included the same two paragraphs describing the attack and its consequences. They differed in their emphasis on the groups that were involved in the storming. In the Republican focus condition, the text briefly acknowledged the involvement of extremist militias and white supremacist groups but focused on the heavy involvement of regular Republican voters as well as Republican party leaders. In the fringe focus condition, the text briefly acknowledged the role of regular Republican voters and leaders but focused on the involvement of extremist fringe groups, anti-government militias and white supremacists. The wording for each condition can be found in S9 Table in S1 File.

Subjective temporal distance was once again our primary variable of interest (1 = *Feels very recent* to 100 = *Feels very long ago*.). As in Studies 1 and 2, we excluded a second subjective distance item on the advice of a reviewer; and results with the two-item measure, which reveal very similar patterns, can be found in the supporting information.

Current relevance was captured using three measures. The first, representing relevance to the Republican Party in the present, included three items rated on 7-point Likert scales ("How relevant do you consider this event to be to the Republican party **in the present**?", "To what extent do you believe this event reflects the current state of the Republican party?", and "How important is this event for your current opinion of the Republican party?"). The second included the same three items but pertaining to US politics, and the third used the same items in reference to former Republican president Donald Trump. Because all three measures showed good reliability when combined ($\alpha = .95$) we decided to use one aggregate current relevance item for our analyses as in past studies.

Participants responded to a memory check where they were asked to recall which pieces of information were presented in the description of the event. "The Republican voters making up the crowd" and "The Republican lawmakers involved in the event" matched up to the Republican focus condition, while "The different extremist groups participating in the assault", "The white supremacist element in the assault" and "The involvement of the Oath Keepers and Three Percenters" matched up to the fringe focus condition. See supporting information for additional measures on perceptions of the Capitol riot.

## Results and discussion

Because all conditions discussed a negative event for the Republican party, participants were already naturally divided into supporters (Republicans) and opponents (Democrats). In line with our hypotheses, voter affiliation and focus condition were the main independent variables of interest.

**Memory checks.** Our memory checks assessed whether people recalled the relevant groups mentioned in the two focus conditions (see Table 7). Overall, a majority of participants recalled mention of Republican voters and lawmakers respectively in the Republican focus condition, and a majority of participants recalled the involvement of extremist groups, white supremacists, and Oath Keepers/Three Percenters respectively in the fringe focus condition. Notably, because both types of groups were mentioned in both passages, the involvement Republican and fringe actors was still recalled outside of their respective focus conditions, but

**Table 7. Cell counts and percentages for groups recalled in Capitol riot description by focus condition.**

|  | Republican Focus (n = 359) | Fringe Focus (n = 348) |
|---|---|---|
| **Extremist Groups** | 186 (51.8%) | 313 (89.9%) |
| **White Supremacists** | 192 (53.5%) | 225 (64.7%) |
| **Oath Keepers & Three Percenters** | 7 (2%) | 307 (88.2%) |
| **Republican Voters** | 299 (83.3%) | 54 (15.5%) |
| **Republican Lawmakers** | 310 (86.3%) | 62 (17.8%) |

Table reports total number of participants in each condition that recalled the group as being mentioned in our description of the Capitol riot along with the percentage this represents.

by a smaller percentage of participants. (Note that the Oath Keepers and Three Percenters were not mentioned by name in the Republican condition, which is likely why they were recalled at much lower rates than extremist and white supremacist groups).

**Subjective temporal distance.** As predicted, Republicans saw that Capitol attack as significantly farther away than Democrats, $F(1, 701) = 62.66$, $p < .001$, $\eta_p^2 = .082$. There was also a marginal main effect of focus condition, such that the attack seemed somewhat closer in the Republican focus condition than the fringe condition, $F(1, 701) = 3.51$, $p = .061$, $\eta_p^2 = .005$. Contrary to our predictions, however, there was no interaction between voter affiliation and focus condition, $F(1, 701) = 0.12$, $p = .726$, $\eta_p^2 < .001$. Estimated marginal means and standard errors can be found in Table 8. The lack of interaction is inconsistent with our prediction that Republican participants would distance the Capitol attack more when it was closely tied to the Republican party (vs a fringe), whereas Democrats might view it as closer when it was more linked to Republicans. As noted in the prior section, it may be that the overwhelming attribution for blame to the fringe (and other sources) across both focus conditions weakened the impact of any perceived responsibility of Republican voters and leaders and diminished the chance of detecting a condition effect. It may also be that both attributions to Republicans and to a distasteful fringe are equally threatening to Republicans and motivate the same desire to distance regardless of the frame.

**Current relevance.** As predicted, Democrats saw the Capitol attack as more relevant than Republicans, $F(1, 703) = 822.99$, $p < .001$, $\eta_p^2 = .539$. As well, the attack seemed more relevant in the Republican focus (vs fringe focus) condition, $F(1, 703) = 7.01$, $p = .008$, $\eta_p^2 = .010$. Again, inconsistent with predictions, there was no interaction between voter affiliation and focus condition, $F(1, 703) = 1.06$, $p = .303$, $\eta_p^2 = .002$. Estimated marginal means and standard errors can be found in Table 8.

Although our predictions for the moderating effect of focus condition were not supported, results provide a strong preregistered replication of the key phenomena we have been examining throughout this paper. Republicans, for the whom the Capitol storming reflects most negatively on their own political ingroup, placed this objectively still-recent event in the relatively distant subjective past, and viewed this event as significantly less currently relevant relative to Democrats.

**Table 8. Estimated marginal means and standard errors by cell for subjective time and current relevance.**

|  | Republicans | | | Democrats | | | Total | |
|---|---|---|---|---|---|---|---|---|
|  | Republican Focus | Fringe Focus | Total | Republican Focus | Fringe Focus | Total | Republican Focus | Fringe Focus |
| Subjective Time | 43.85 (2.02) | 46.95 (2.18) | 45.40 (1.48) | 27.03 (2.01) | 31.56 (1.93) | 29.29 (1.39) | 35.44 (1.42) | 39.26 (1.46) |
| Current Relevance | 3.50 (0.09) | 3.15 (0.10) | 3.33 (0.07) | 6.10 (0.09) | 5.95 (0.09) | 6.03 (0.06) | 4.80 (0.07) | 4.55 (0.07) |

Numbers in parentheses correspond to standard errors.

The very strong effects of partisanship may have overwhelmed the subtle manipulation of focus, especially since both mainstream and fringe groups were mentioned in both conditions and arguably the participation of both mainstream and fringe groups reflect poorly on Republicans.

## General discussion

"*The election ended a long time ago in one of the biggest Electoral College victories in history. It's now time to move on and 'Make America Great Again'.*"

> • *Trump Administration response to CIA allegations of foreign interference in the 2016 U.S. election, one month after the vote.*

The relevance of past events to a candidate's ability to lead is often ambiguous at best. Whether or not a decade-old gaffe is considered to be ancient history or recent news depends on whether the past mistake belongs to a candidate we hope to see win or lose—or, perhaps, how that gaffe is communicated to us.

Over multiple studies, political incidents, and time periods, we present evidence demonstrating that people's subjective feeling of time varies according to whether or not a past event is helpful or harmful to the conclusion they wish to draw in the present. People perceive their supported candidate's successes and an opposing candidate's failures to be close in time and relevant to judgements in the present, while viewing their favored candidate's equidistant failures and opposing candidate's successes are a part of the distant past and irrelevant to current appraisals. This pattern occurred for real political leaders and fabricated ones, and for events only several months old to events decades in the past. These findings also extend past research on political scandals [1–3] and offer insight into the cognitive and motivational mechanisms that may characterize how voters think about such incidents.

Moreover, although time perceptions are highly elastic on the basis of partisanship, they can also be experimentally altered by framing a past event as recent or distant—even in the presence of the objective date. Study 4 demonstrated the causal effect of subjective distance: when people were experimentally induced to see a political scandal as temporally remote (vs recent) they judged the marital infidelity as less currently relevant, and evaluated the candidate more favorably and even as more worthy of their vote. These effects demonstrate the potential influence of 'minor' descriptors in print, such as the Trump administration's response to CIA allegations [34]. The authors note that while the overall effect of the temporal distance manipulation was relatively small, small effects have impact when applied over large populations, such as the voting population of the United States. This effect may extend to any medium in which an event is described and framed such as newsprint or blogs. In application these findings have implications for any candidate aspiring to office who hopes to weather public scrutiny of their past record. Indeed, future research could examine in greater detail how and when subjective distancing may help a candidate relegate past misdeeds to a former self, and when such strategies could backfire. to fully demonstrate the relationship.

Theoretically, these results present a previously unexplored use of subjective temporal distance: subjective time can be wielded both to defend and attack others' character. Further, they partially support the hypothesis that political beliefs can shape people's representations of how autobiographical history contributes to current identity. In the extreme, this work suggests that partisans can find themselves viewing two different versions of history. It is not unreasonable to think that something that happened long ago might have limited relevance for the present, so if people are largely unaware of how their perceptions of time are shifting they may have little cognizance of how the conclusions they draw are systematically skewed. If people's

sense of what events seem like yesterday versus far away are shaped by political motivations, partisan opponents might profoundly disagree about the relative importance of a past political scandal and have little ability to understand why their opponent sees it so differently. However, it is important to note that the current findings cannot conclusively determine the mechanism leading to these shifts in subjective time. For instance, we cannot determine with certainty whether the process is subtle and automatic or an explicit distortion–indeed, although people may not always be aware of how their experience of time shifts, they may well sometimes use distancing intentionally and strategically, as when political communicators publicly relegate missteps to the distant past. Determining more precisely the nature of the underlying processes behind shifting perceptions of time would be a beneficial avenue for future research.

How politicians and political groups intuit the relevance of time, and how their personal statements on past actions (e.g., Donald Trump on Mike Pence's Iraq vote: "**[It was] a long time ago. And he voted that way and they were also misled. A lot of information was given to people.**" [35]) affect people's perceptions has yet to be shown. It may be that voters are less inclined to believe statements directly from a candidate than from a spokesperson or media outlet, or that such statements move a leaning undecided to take one further step toward open support. That said, these results showcase the impact of time and the importance of how an event is framed.

## Limitations and future directions

**Measurement.** Because our measures were self-report, we cannot say how conscious people were of their systematic shifts in subjective time. Since psychological judgments of time are often fluid and constructed, participants may have been largely unaware of how they truncated or expanded time in ways that supported their desired conclusions about political actors and events. On the other hand, it is possible that people alter their judgments of time in a more deliberate act of self-presentation, making strategic claims about time to intentionally shift the narrative.

With regard to the interpretation of subjective time, it is also important to consider the possibility that the measure was used to express something other than people's genuine subjective perceptions of time. Researchers have pointed to the methodological challenge of *response substitution* that can affect survey research in hidden ways [36]. Substitution occurs when participants wish to express attitudes or beliefs that have not yet been asked about and use a seemingly unrelated question to express this view. For example, if a customer is given poor *service* at a restaurant but the feedback survey only asks about the *food*, they will give the food a poor rating, regardless of its actual quality, as a means of expressing their negative experience [36]. In the current studies, if participants were asked about subjective time before they have a chance to express their attitudes towards a past scandal, then they could conceivably use the opportunity to report the scandal's timing as closer or farther as a means of expressing the evaluation they wish to convey. By this logic, someone might opt to say an event "feels like ancient history" not because it actually seems more distant in time but as a means of expressing "it's not a big deal." Indeed, subjective distance measures that incorporate a metaphor like "ancient history" could be especially susceptible to such a concern–people may understand "ancient history" as a dismissal of current relevance, whereas measures simply asking whether an incident seems recent or long ago may be less loaded in this manner.

Although the concern about response substitution is important, we took several steps to assess this possibility and argue that it is not a likely alternative interpretation of the current findings. First, although in some studies we included the more loaded "ancient history" metaphor as a scale endpoint, it was always asked second after a more subtle measure ("*feels very recent/long ago*"). Results are largely identical when analyses are limited to the more subtle

measure of subjective time (which was always asked first). To offer the most conservative test of our hypotheses, we report results in the manuscript using only the subtle measure of time perception and report results of the two-item scale including "ancient history" in the supporting materials. Second, research on response substitution recommends considering question–specifically, including direct evaluation questions that allow people to express their views first, in order to reduce the likelihood that people will respond expressively on proxy items [36].

Although our studies were not designed with this order concern in mind, when reviewing our study methods we note that order of question did vary across studies in ways that help us address the question of response substitution. Specifically, in Studies 1, 2, 4, and 5 the subjective-time questions were asked first, but in Study 3 (and all precursor studies reported in supporting materials), subjective time questions were asked *after* other items that allowed people to express their evaluations and attitudes about the incident. Descriptively, the patterns observed were the same regardless of whether the temporal distance items came immediately or later in the survey. Therefore we argue that subjective distance is more likely to be measuring people's highly elastic psychological sense of time [4, 5] than simply being a proxy for respondents to express their more general approval or disapproval.

**Generalizability.** Despite the wide variety of contexts we examined across 8 studies in total (including those in supporting materials), our focus was always on politics in the United States and Canada, two very WEIRD countries with extensive cultural overlap [37, 38]. Some psychological constructs, including some basic motivated reasoning and political impulses, do appear to generalize across cultures [39, 40]. But cross-cultural generalizability is something to be tested, not assumed. It is possible that the motivated shifts in temporal perception documented here would not appear in cultures that perceive time differently [41, 42]. The focus on political contexts in particular also limits the scope of our findings, though studies were intentionally designed to test this domain and extend past reseat examining motivated shifts in time in the service of self- or ingroup-protection [4, 7, 43, 44]. Future research could examine other contexts where subjective time is shifted to either impugn or excuse the acts of others–applications could range from criminal justice to "cancel culture".

## Conclusion

Although it is sometimes said that time heals all wounds, how people subjectively experience time may differ meaningfully from chronological or calendar time and may play an important role in how people connect the dots from past actions to present judgments of character. The current research demonstrates that political motivation to protect co-partisans or attack opponents can profoundly shift the perception of time itself. In turn, perceiving an event as more remote—regardless of calendar time—can make it appear less relevant to the present. In the context of the stark and rising political polarization currently in the USA [45, 46], these results reveal yet another way that partisans may "talk past one another" by judging something as seemingly innocuous and apolitical as the passage of time in different ways that may lead their conclusions about even momentous past events to diverge sharply. With strong enough motivation (in this case, fueled by partisanship), psychological time travel is possible: a major long-past event can quickly be returned to the present or relegated to a remote, long-closed chapter of the past, leading partisans to polarize even in their judgments of who deserves credit for, or redemption from, their past glories and sins.

## Supporting information

**S1 File.**
(DOCX)

## Author Contributions

**Conceptualization:** Andrew J. Dawson, Scott A. Leith, Cindy L. P. Ward, Sarah Williams, Anne E. Wilson.

**Data curation:** Andrew J. Dawson, Scott A. Leith, Cindy L. P. Ward, Sarah Williams.

**Formal analysis:** Andrew J. Dawson, Scott A. Leith.

**Funding acquisition:** Anne E. Wilson.

**Investigation:** Andrew J. Dawson, Scott A. Leith, Cindy L. P. Ward, Sarah Williams, Anne E. Wilson.

**Methodology:** Andrew J. Dawson, Scott A. Leith, Cindy L. P. Ward, Sarah Williams, Anne E. Wilson.

**Project administration:** Andrew J. Dawson, Scott A. Leith, Cindy L. P. Ward, Sarah Williams, Anne E. Wilson.

**Resources:** Anne E. Wilson.

**Supervision:** Anne E. Wilson.

**Visualization:** Andrew J. Dawson.

**Writing – original draft:** Andrew J. Dawson, Scott A. Leith, Cindy L. P. Ward, Sarah Williams.

**Writing – review & editing:** Andrew J. Dawson, Anne E. Wilson.

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
