## [Decision Letter · Decision Letter 0]

14 Jun 2022

PONE-D-22-13208Far away or yesterday? Shifting perceptions of time for political endsPLOS ONE

Dear Dr. Dawson,

Thank you for submitting your manuscript to PLOS ONE. After careful consideration, we feel that it has merit but does not fully meet PLOS ONE’s publication criteria as it currently stands. Therefore, we invite you to submit a revised version of the manuscript that addresses the points raised during the review process.

We were fortunate to have three experts review this paper. In addition to my own, independent, read, the entire team finds the topic quite interesting, the main finding about time perception important, but feel that the paper could be improved a fair bit. To that end, I’m considering this a “major revision” meaning that it will go back to the reviewers upon re-submission. There is a fair bit of work to do here. I will highlight some of the key concerns that the review team shares, but I encourage you to carefully read all the reviews and reply to the comments as appropriate.

First, there is consensus that the paper is too long given the contribution being made. Notably, all the reviewers agree that the evidence supporting the claim about time perception is (mostly) solid, but that the rest of the claims aren’t well supported by the evidence. To the end, I very much like the suggestion of R2 to pare back the paper to focus just on the time perception finding, and to limit the rest of the results to the supplementary materials. Alternatively, you could follow the suggestion of R1 to re-run Study 4 as a pre-registered study. Positive results there would help alleviate the concerns raised by the review team. This latter option is obviously more difficult, and so I leave it to you to decide which path to pursue.

Regardless of which option you take, I suggest paring back on the analyses reported and moving much of the speculative parts to a supplement. Doing so will streamline the paper considerably.

Empirically, my biggest concerns are the ones raised by R1 about the measurement instrument (“ancient history” is a very loaded anchor point) and response substitution (Gal and Rucker 2011). To summarize the latter, the concern is that participants aren’t truly indicating a belief about time perception, but rather are indicating their general feelings towards the information provided. The paper I cite above provides some very clear methodological solutions to this problem that I would ask that the authors address. You can do that empirically with a new study (the best option) or rhetorically if you believe you have enough evidence to counter this concern.

On the whole, I very much like this paper and the findings and hope that the authors incorporate this feedback and improve the paper. I look forward to reading the revision.

Reference: Gal, D., & Rucker, D. D. (2011). Answering the unasked question: Response substitution in consumer surveys. *Journal of Marketing Research*, *48*(1), 185-195.

We look forward to receiving your revised manuscript.

Kind regards,

Jeff Galak, PhD

Academic Editor

PLOS ONE

Journal Requirements:

2. We noted in your submission details that a portion of your manuscript may have been presented or published elsewhere. 

[The three precursor studies included in the supporting information use data from a prior set of studies on implicit theories of change. In theses studies, measures of subjective time were included after the primary study materials on a separate page. The original studies were published in the paper referenced below. This paper has been included as a related manuscript.

Leith SA, Ward CL, Giacomin M, Landau ES, Ehrlinger J, Wilson AE. Changing theories of change: strategic shifting in implicit theory endorsement. J Pers Soc Psychol. 2014 Oct;107(4):597

Aside from basic descriptive statistics about the samples, all analyses reported on the precursor studies in the supporting information are novel and were not included in Leith et al.] 

Please clarify whether this publication was peer-reviewed and formally published. If this work was previously peer-reviewed and published, in the cover letter please provide the reason that this work does not constitute dual publication and should be included in the current manuscript.

Reviewers' comments:

Reviewer's Responses to Questions

**Comments to the Author**

1. Is the manuscript technically sound, and do the data support the conclusions?

Reviewer #1: Partly

Reviewer #2: Yes

Reviewer #3: Yes

2. Has the statistical analysis been performed appropriately and rigorously? 

Reviewer #1: Yes

Reviewer #2: Yes

Reviewer #3: Yes

3. Have the authors made all data underlying the findings in their manuscript fully available?

Reviewer #1: Yes

Reviewer #2: Yes

Reviewer #3: Yes

4. Is the manuscript presented in an intelligible fashion and written in standard English?

Reviewer #1: Yes

Reviewer #2: Yes

Reviewer #3: Yes

5. Review Comments to the Author

Reviewer #1: This paper proposes that people will report past political events as being subjectively more or less distant to the degree that the event is positive or negative from their partisan point of view. I found the evidence presented quite convincing of the basic effect. However, the paper also makes stronger causal claims, that partisanship distorts actual perceptions of time. I thought the evidence was somewhat incomplete on that dimension.

I think the strongest alternative explanation is “response substitution” or “attribute substitution”: people are choosing to answer a different question than the one being asked. Specifically, when asked about the timing of something that challenges their political views, they use that question to instead express their views about whether it’s a big deal. Of course, this presupposes that people have the proposed association in their minds (long ago = no big deal). The difference in this alternative account is that they don’t actually believe that it was long ago – they just say that to express their view that it’s not a big deal.

From this perspective, there are some key empirical limitations in the paper. First, the key DV is measured using two questions, one of which involves the term “ancient history.” This is a loaded term, often used when political acts are being minimized. At a minimum, it would be helpful to test how robust the results are to only using the more objectively measured variable (1 = Feels very recent; 100 = Feels very long ago). From the same perspective, it would be better to ask the key questions first (i.e., before measuring political identity), at least in some studies or some conditions, because asking about identity first may prompt people to think of the time-perception question as an expression of their identity.

In Study 2, Clinton supporters paradoxically rated both a higher length of the Trump presidency and a longer time until the 2020 election. One interpretation of this is that people associate painful events with the slow subjective passage of time. For a person who finds the Trump presidency highly aversive, much like a root canal, it will feel like it has been going on for a long time and still has a long time to go, or even if it doesn’t actually feel that way, that may be a salient metaphor for expressing their disapproval. I think this could be consistent with either the proposed process or with the attribute substitution alternative. However, I do think these results are inconsistent with a version of the proposed process in which people have a single distorted timeline in their minds (as opposed to the distortion occurring in the context of specific questions).

Was the discussion of manipulating external framing (on p. 50) referring to Study 4? I had a concern about the manipulation in that study. The text says “Mr. Bosch was reported as having an affair outside of his marriage in 2008” and later refers to a 2008 report again. Participants may have interpreted this as meaning that 2008 refers to when the report occurred, not when the affair happened, and then attributed entirely different dates to the affair based on the manipulation. Therefore, the manipulation may represent an actual informational difference, rather than mere framing.

I appreciated the open-ness regarding the limitations of Study 5. Given that this was the only pre-registered study, and that multiple aspects of the study did not work as planned to create the conditions in which the hypotheses could be tested (i.e., the focusing manipulation seemingly did not shift responsibility as intended and the verbal fusion scale had a problem), this study did not add as much to the paper. The interaction with identity is consistent with the hypothesis, but since this only occurred with measure not designated as the primary analysis it’s not as conclusive. (As an aside, you may be interested in Chen & Urminsky 2019, which presents a measure of political identity that relates to political attitudes and self-reported behaviors).

Overall, I think the current paper makes a strong case that people report events challenging their political views as subjectively more distant. I think the paper makes a weaker case that this represents an actual perception that impacts subsequent attitudes (as opposed to being an outcome of other attitudes or outright attribute substitution). I think this could be addressed by discussing these limitations and toning down some of the claims in the GD. Alternatively, I think re-running a pre-registered version of Study 4 with more precise language and some comprehension checks could address these issues. If the latter strategy is taken, I think it could be useful to also include measures of political identity and test the moderation found in Study 5.

Minor points:

The scenario quoted in Study 3 only shows the negative version.

It would be useful to include the survey instruments in the OSF archive.

Reviewer #2: This paper reports findings from a series of 5 experimental studies. The studies set out to assess whether motivated reasoning affects subjective assessments of how long ago a politician’s (mostly negative) behavior occurred. The design of the experiments is solid and the replication of core findings across studies is commendable. The core finding—that people tend to view negative behavior from political opponents as more recent, but identical information about an ally as distant—is a useful contribution to our understanding of how people process political information.

The primary limitation of the paper is that far too much extraneous analysis is presented. This leads to bloat that distracts from the core contribution of the paper. The additional analysis is also undertheorized and sometimes feels like it is coming out of nowhere. I would encourage the authors to pare out large swaths of what is presented and either simply exclude this additional material or confine it to an appendix. Consider Study 5. This is the fifth study on the theme presented in the paper. My read is that one contribution is to simply use a very high salience, recent event (January 6 insurrection). The other is that the experiment varies whether the events were framed as being precipitated by “fringe groups” or the Republican Party more generally. If so, it seems like this section should be concise, noting similarities in findings to Studies 1-4 and describing what the “fringe/mainstream” treatment adds. Instead the section of the paper on this study stretches from page 37 to page 49.

Much of this bloat is driven by exploration of how the treatments affected an array of outcome variables beyond subjective perceptions of how long ago an event happened. At various points, stray potential moderators like media consumption are brought in. Table 7 alone takes up almost a page exploring how the treatments affected (or didn’t affect) perceptions regarding who was responsible for the riots. Table 8 pursues this further reporting marginal means by condition for each of 9 potential entities who respondents might view as responsible. In a sense this seems like a manipulation check (did focusing on extremist groups lead, say, Republican respondents to view the events as less about “their group’s” behavior?) If so, this analysis should not be allotted multiple pages of the paper, while discussion of the core question gets half a page (p. 45). Skip to the analysis that matters and note that the null effect of the focus treatment here may be a product of that treatment failing to meaningfully shift how people thought about who “did it.” Refer readers to an Appendix for more detail.

I am sympathetic to the authors’ impulse to try to use all of the data and fit a bunch of pieces together. My read is that these efforts raise far more questions than they answer. For example, I understand that the authors would like to show that subjective temporal distance perceptions affect vote choices. However, these data do not allow them to demonstrate that. Consider the path analysis reported in Figure 4. This model rests on critical *assumptions* about the chain of causality. I am skeptical that the model proposed is actually how things work. Why should we believe that group attachment to a political figure affects subjective temporal distance, which in turn affects perceived relevance, which in turn affects assessments about the political figure’s morality (etc), which in turn affects vote intentions? It seems at least as plausible to posit that people like candidates/figures from their preferred parties and are inclined to vote for them. When confronted by bad behavior from the past, they may rationalize their vote intentions by reasoning that the event isn’t really such a big deal and, after all, it happened quite a while ago, didn’t it? I would suggest cutting the path analysis from the paper as it implies that the correlations between post-treatment variables can be used to shed light on the chain of causality. They cannot.

In short, this could be a strong, focused 20-25 page paper that offers new insights into how motivated reasoning affects how people respond to information about instances of past wrongdoing. Existing work (e.g., Pereira and Waterbury 2019; Doherty, Dowling, Miller 2014) shows that the effects of scandals fade with the passage of time. This paper offers new insights into how partisans may reason about past events like this. If the authors want to show that subjective assessments of temporal distance are correlated with other judgments (e.g., vote intent; perceived relevance to current evaluations), they certainly can do so. However, they should be agnostic about the chain of causality.

*** Additional notes

The revisions I suggest above mostly involve cuts. I would suggest the authors use some of this “room” to provide a bit more detail about the treatments used in Studies 1-3. Where did the text of these treatments come from? Should we be concerned that the text of the treatments is so different across conditions?

Similarly, I’d like to see a bit more about sample restrictions. E.g., on page 8 the authors refer to participants “whose bias could not be reasonably guessed.” How did the authors try to “guess”? And exclusions tied to attention checks should consistently explain what the attention check was. In some cases the exclusions are substantial (e.g., 623 get pared to 327 in Study 2). The reader needs to know how many lost cases each exclusion results in.

Tables tend to be easier to interpret if their title and/or notes are clear about what variable is being analyzed (e.g., Table 1)

Are the power tests reported the right ones? They seem to pertain to direct effects, but the estimates of interest are almost all (all?) tied to interactions.

Cut the snarky first sentence from the General Discussion.

Reviewer #3: This paper investigates a phenomenon that can be related to the recent literature on motivated beliefs. People perceive bad past events more in the past than good events. Specifically, they think that bad behaviors of supported politicians are more in the past than bad behaviors of not supported politicians. The study uses well-powered surveys conducted with MTurk. The dependent variable is based on two qualitative questions on how much in the past people think certain events are. The main effect (an interaction of whether one favors the politician and whether the event is positive or not) was found in five independent studies. The individual studies vary in whether they study real or fictitious politicians, in the events and some of the studies introduced a manipulation.

The research question is interesting and given that the effect could be shown in five independent experiments it appears as robust. The phenomenon is relevant because it identifies a mechanism for a bias in favor of preferred candidates. However, it is not clear how a debiasing strategy could look like. The analysis is comprehensive and makes clear what is in the focus and what has to be considered as more explorative.

Specific comments

1. The structure follows the standard of psychology papers presenting the studies sequentially. This structure is less common in other fields such as economics. Since the studies differ in the variables that are collected, this is not a bad structure. Nevertheless, I recommend adding more integrating information. In particular, a summary table reporting all studies would be helpful.

2. Tables and figures are not self-contained. I recommend adding more information. Further, even if this is irrelevant for the published paper, it would be helpful for the referees if there were no page breaks within tables.

3. Study 2. In order to classify people as Clinton or Trump supporter, people were asked to report whom they voted for, and those who were “very dissatisfied” with their vote at present were excluded. I wonder why they did not directly ask for the current support for these candidates. Excluding always reduces the number of observations and, if not pre-specified, adds arbitrariness in the analysis.

4. Page 19, “we did not necessarily predict the same interaction”. This formulation feels strange. I believe the authors that their focus variable was the subjective distance from the 2016 election, and that the other variables were more explorative. Nevertheless, I would explicitly consider this as explorative analyses.

5. Page 21, analysis of the post-election media exposure × voter affiliation regression. The report of the regression coefficients in the text are problematic, because the main effects depend on the how the other variable is defined (omitted category). I would prefer to see the regression information in a table and that the discussion of the regression more quickly starts with the slope analysis.

6. Obvious malingerers were excluded in study 3. Were similar exclusions possible in the other studies.

Minor issues

7. In Table 1 valence is negatively coded, i.e. bad is positive. This is confusing, in particular because it seems different in Table 4.

8. I recommend putting the middle category into the middle in tables. For example, in table 5, the comparisons would be more natural if the order would be Close – Control – Distant and Supporter – Neutral – Opponent.

9. In the beginning of the general discussion, some statements were stroked though. What is the status of these statements? Anyhow, the beginning of this chapter is not so convincing.

Typo

10. Page 6: Replace “In an effort to be maximally transparent, we include three studies, from a prior paper investigating implicit theories of change [14], are included in the supporting information to this paper.” by “In an effort to be maximally transparent, three studies from a prior paper investigating implicit theories of change [14] are included in the supporting information to this paper.”

11. Page 25, “supporting informaiton”

6. PLOS authors have the option to publish the peer review history of their article (what does this mean?). If published, this will include your full peer review and any attached files.

Reviewer #1: **Yes: **Oleg Urminsky

Reviewer #2: No

Reviewer #3: No

---

## [Author Response · Author response to Decision Letter 0]

3 Aug 2022

Copied from response letter:

Dear Jeff Galak and the review team,

Thank you for your comments. We appreciate your thoughtful feedback about both the manuscript’s strengths and limitations, and have carefully considered all of your and reviewers’ constructive critiques and suggestions. We believe that this process has resulted in a substantially improved (and much shorter!) paper. In accordance with PLOS One’s review standards, we have made particular effort to focus only on the key findings and to ensure that our conclusions do not overclaim the strengths of the evidence. In the letter below we include the text of your and reviewer’s feedback in italics followed by our responses immediately after each point. 

Editor’s comments

Thank you for submitting your manuscript to PLOS ONE. After careful consideration, we feel that it has merit but does not fully meet PLOS ONE’s publication criteria as it currently stands. Therefore, we invite you to submit a revised version of the manuscript that addresses the points raised during the review process.

We were fortunate to have three experts review this paper. In addition to my own, independent, read, the entire team finds the topic quite interesting, the main finding about time perception important, but feel that the paper could be improved a fair bit. To that end, I’m considering this a “major revision” meaning that it will go back to the reviewers upon re-submission. There is a fair bit of work to do here. I will highlight some of the key concerns that the review team shares, but I encourage you to carefully read all the reviews and reply to the comments as appropriate.

First, there is consensus that the paper is too long given the contribution being made. Notably, all the reviewers agree that the evidence supporting the claim about time perception is (mostly) solid, but that the rest of the claims aren’t well supported by the evidence. To the end, I very much like the suggestion of R2 to pare back the paper to focus just on the time perception finding, and to limit the rest of the results to the supplementary materials. Alternatively, you could follow the suggestion of R1 to re-run Study 4 as a pre-registered study. Positive results there would help alleviate the concerns raised by the review team. This latter option is obviously more difficult, and so I leave it to you to decide which path to pursue.

Regardless of which option you take, I suggest paring back on the analyses reported and moving much of the speculative parts to a supplement. Doing so will streamline the paper considerably.

We appreciate your and the reviewer’s thoughtful, well-aimed criticism and constructive suggestions for improvement. We address these points in detail in reviewer comments below, but in summary: 1) we agree the paper was too long! We have pared it down by 14 pages in the ways you have suggested. Some of the analyses we cut are reported in the supporting materials for readers’ additional interest but we have been careful not to include interpretations that overclaim results in either the manuscript or supporting materials. 

2) We focus analyses (studies 1,2,3,5) more to highlight that subjective time perception effects are the main contribution, and eliminate path models since their additional evidence value is weak. Both for transparency and added informational value, we do report perceived current relevance and a couple of other dependent variables in later studies, but now simply report their effects by condition and don’t suggest causal connections as we previously did in path models. For example, people’s conclusions about current relevance shift by condition in the same manner as subjective time perception (viewing opponents past misdeeds and ingroups’ successes as more presently relevant than the reverse); and offers additional information about how people assess the psychological statute of limitations of past events. However, path models suggesting that subjective time mediates relevance judgments are eliminated since multiple causal directions are plausible and we agree that our interpretations sometimes went beyond the evidence. 

3) Study 4 is the only study that manipulates the subjective framing of time rather than examining subjective time perception as a dependent variable. As a result, it can offer some preliminary evidence of the causal role subjective time when people weigh past misdeeds in their judgments of someone in the present. We drop the path models in Study 4 for the same reasons noted previously. We recognize that R1 had concerns about a possible alternative explanation for Study 4 and we believe we can address that reasonably well by clarifying methods (see our detailed response to R1 below). We opted not to run a preregistered replication of Study 4, in part because the theoretical contribution (demonstrating that subjective time can be manipulated and causally influence judgments of others) is not new and has been previously demonstrated in other contexts (e.g., close relationships; Cortes et al., 2018), though we believe it is still valuable to test it in the context of political scandals using a verbal framing technique (as we do) that might mirror a real-world description of such political events. We also argue that Study 4 helps clarify why it matters that people shift their perceptions of time for political events by directly testing its causal effect on judgments. Instead of running an additional study, we clarify the method, contextualize the purpose of the study and the contribution it makes in the context of past research, and are more careful in the conclusions we draw. 

Empirically, my biggest concerns are the ones raised by R1 about the measurement instrument (“ancient history” is a very loaded anchor point) and response substitution (Gal and Rucker 2011). To summarize the latter, the concern is that participants aren’t truly indicating a belief about time perception, but rather are indicating their general feelings towards the information provided. The paper I cite above provides some very clear methodological solutions to this problem that I would ask that the authors address. You can do that empirically with a new study (the best option) or rhetorically if you believe you have enough evidence to counter this concern.

On the whole, I very much like this paper and the findings and hope that the authors incorporate this feedback and improve the paper. I look forward to reading the revision.

Reference: Gal, D., & Rucker, D. D. (2011). Answering the unasked question: Response substitution in consumer surveys. Journal of Marketing Research, 48(1), 185-195.

Thanks to R1 for raising this methodological concern and thank you for bringing this very interesting paper to our attention! Although we had been intuitively aware of the possibility of response substitution in some survey research, we had not considered it here, and we were not familiar with this paper which documents the phenomenon so nicely and offers helpful solutions. See response to R1 for a full response, but in short, we address the issue in two ways; 1) with additional data analysis (the subjective time item with the loaded “ancient history” endpoint was always included second and patterns do not meaningfully change if we only analyze the first, much more subtle measure of subjective time) and 2) by examining and clarifying the order of item presentation across studies (although subjective distance sometimes comes first where it could be most vulnerable to response substitution, in other studies it is asked after people have a chance to overtly evaluate the incidents in question. Descriptively order does not make a difference in study results. Although we recognize that assessing this pattern across studies is not as compelling as experimentally manipulating order, we believe that taken together, the variation in order across studies is consistent with recommendations by Gal and Rucker and allay concerns that subjective time perception is merely a way of expressing valence evaluations. 

As an aside, thanks again to you and R1 for sharing this insight. The Gal and Rucker (2011) paper has now shaped the method of another study in our lab in a line of work in which (we believe) response substitution has been a far more significant problem – it was something we had been struggling with and had not previously found a way to address. So you and R1 have improved (at least) two programs of work with one suggestion!

We have reviewed PLOS ONE’s style requirements and ensured we meet these guidelines. 

2. We noted in your submission details that a portion of your manuscript may have been presented or published elsewhere. 

Please clarify whether this publication was peer-reviewed and formally published. If this work was previously peer-reviewed and published, in the cover letter please provide the reason that this work does not constitute dual publication and should be included in the current manuscript.

No data from any of the studies in the main manuscript have been published elsewhere. We make reference to three studies (which we describe as “precursor” studies) that were included in a different peer-reviewed publication. These three studies were conducted with a different primary goal (to examine shifts in people’s implicit theories of change or stability). However, subjective distance measures were included near the end of those studies for exploratory purposes. Because subjective distance data were not relevant to the hypotheses in Leith et al (2014), none of the subjective distance measures were reported in the previously published manuscript: 

Leith SA, Ward CL, Giacomin M, Landau ES, Ehrlinger J, Wilson AE. Changing theories of change: strategic shifting in implicit theory endorsement. J Pers Soc Psychol. 2014 Oct;107(4):597

Exploratory analyses of subjective distancing in those earlier studies prompted us to formulate a priori hypotheses and conduct the five studies in the current manuscript. Because the main portion of those past studies was reported elsewhere and subjective time analyses were exploratory, we opted not to include them in the current manuscript, but for transparency and to contribute to documenting the generalizability of the subjective time effect across even more political contexts, we chose to report those subjective distance findings in supporting materials. Aside from basic descriptive statistics about the samples, all analyses reported for the precursor studies in the supporting information are novel and were not previously reported in Leith et al. We aimed to be as transparent as possible about the prior publication and emphasize that even in the supporting materials there is no duplication of findings reported in Leith et al. 

We also note that by reporting the three precursor studies we empty the “file drawer” - that is, between the main manuscript and supplementary document, all studies conducted in our lab that have tested the current hypotheses have been reported. 

Finally, the data has been used in previous conference presentations, listed below.

Dawson, A. J., Wilson, A. E., Williams, S., & Leith, S. A. (2021, February) Far away or yesterday? Offensive and defensive temporal distance strategies in politics. Talk presented at the Society of Personality and Social Psychology 2021 virtual convention.

Dawson, A. J., Wilson, A. E., Williams, S., & Leith, S. A. (2022, February). The incredibly old, very far away, not even that relevant, attempted insurrection: Shifting perceptions of time for political ends. Talk presented at the Society of Personality and Social Psychology 2022 convention in virtual format. 72 submissions accepted out of 1900.

Wilson, A. E., Williams, S., & Dawson, A.J. (2021, June) Tainted or transformed: Reckoning with the past. Paper presented at Canadian Psychological Association Convention, Montreal, QC, CAN.

Reviewer 1 Comments

This paper proposes that people will report past political events as being subjectively more or less distant to the degree that the event is positive or negative from their partisan point of view. I found the evidence presented quite convincing of the basic effect. However, the paper also makes stronger causal claims, that partisanship distorts actual perceptions of time. I thought the evidence was somewhat incomplete on that dimension.

I think the strongest alternative explanation is “response substitution” or “attribute substitution”: people are choosing to answer a different question than the one being asked. Specifically, when asked about the timing of something that challenges their political views, they use that question to instead express their views about whether it’s a big deal. Of course, this presupposes that people have the proposed association in their minds (long ago = no big deal). The difference in this alternative account is that they don’t actually believe that it was long ago – they just say that to express their view that it’s not a big deal.

From this perspective, there are some key empirical limitations in the paper. First, the key DV is measured using two questions, one of which involves the term “ancient history.” This is a loaded term, often used when political acts are being minimized. At a minimum, it would be helpful to test how robust the results are to only using the more objectively measured variable (1 = Feels very recent; 100 = Feels very long ago). 

Thank you for this thoughtful and constructive feedback. You note a couple of distinct points here, and we have thought carefully about and aim to address each, both by offering new analyses and methodological clarifications, and by softening our claims and considering alternative interpretations in recognition of PLOS One’s strong emphasis on ensuring that conclusions don’t go beyond what can be claimed from methods and results. 

First, you note that we cannot claim with certainty that what is distorted is participants’ actual perceptions of time. In an expanded section of the discussion we do acknowledge that on the basis of the current research we cannot be sure whether subjective time self-reports really reflect genuine shifts in time perception or something more self-presentational. More broadly, the literature on subjective time makes clear these perceptions (or at least what people report about their experience) are highly elastic, but as with a great many self-report measures, it can be difficult to distinguish genuine experience from other expressive goals. We highlight these alternative explanations so that readers can weigh the conclusions in light of these possibilities. 

Second, you point out that subjective time results could be consistent with “response substitution” - noting both that if people don’t have the chance to express their attitudes about the incidents, they may use the time measure as a proxy for that expression, and also that one of our subjective time items has quite a loaded endpoint of “ancient history” which is a phrase often used when political events are publicly minimized. 

With regard to the item with the “ancient history” endpoint, we can see the argument for concern that this metaphoric item is quite heavy handed and could call our interpretation into question if it were driving the subjective time findings. That item was used in Studies 1, 2, and 5; Study 3 used two Likert-type items without this endpoint (see page 26 with tracked changes for a description in the method section) and Study 4 only used only one slider item without this wording (described on page 35 with tracked changes). The three precursor studies as well only used one item each, none of which contained the “ancient history” endpoint. The fact that subjective time does shift in very similar ways in studies without the loaded endpoint makes a stronger case for the idea that it is perceptions of time that are shifting and not just a metaphoric expression of attitudes. 

However, we do find the issue of the loaded endpoint to be compelling enough that we believe that the subjective time findings are more convincing if reported without that measure included. In each case where the ancient history item was included it always came after a more subtle measure, so we can arguably examine the first measure on its own. As such we opted to consistently report the subjective time analyses omiting the “ancient history” item. Specifically that means that in Studies 1, 2, and 5 we exclude the second item and only analyse subjective time with a one-item measure ranging from (1 = Feels very recent; 100 = Feels very long ago). The results in each study are robust to this change, with all of the key interactions and main effects remaining the same. Specific patterns of simple effects change slightly in two instances: for Clinton supporters, Clinton’s positive vs negative incidents in Study 1 no longer differed (see page 13 with tracked changes) and that for Trump supporters rating the positive and negative passage election in Study 2 (went from marginal to non-significant, see page 20 with tracked changes).

We also include the original two-item analyses in the supporting materials. Please note that we found it a bit awkward to clarify why we included the “ancient history” item in several studies but then omitted it from analysis; therefore we refer explicitly to reviewer feedback to help account for this decision (in the methods of the relevant studies) as well as offering a more thorough consideration of the issues in the general discussion. We would be happy to thank the reviewer by name in acknowledgements as well, and would like to inquire if this is permissible. If the reviewers or editor would prefer that we use the original two-item scale (including ancient history) instead we are happy to revise the manuscript accordingly and could report the analyses omitting “ancient history” in the supporting materials instead. 

From the same perspective, it would be better to ask the key questions first (i.e., before measuring political identity), at least in some studies or some conditions, because asking about identity first may prompt people to think of the time-perception question as an expression of their identity.

The issue of response substitution posed by the order of the scales was a key point of theoretical concern when we were revising the paper. We also carefully considered the Gal & Rucker (2011) paper suggested by the editor as offering methods to address and mitigate response substitution. Please also see our response to the editor for additional reflections on this issue. Due to its importance in how we interpret the findings, we devoted a short section in the general discussion to this issue (pages 60-61 with tracked changes, and the key section is excerpted below for your convenience). However, as discussed in the new paragraphs, we believe that our full set of studies can address this concern reasonably well. If subjective distance were the first thing participants are asked to report after reading about political scandals and events, it is indeed possible that they would use those items as a proxy for expressing their reactions to the events. Indeed, Gal and Rucker (2011) suggest altering order of questions as one way to address this problem – if people have a chance to say what they really want to express, they may be less likely to use proxy items for expressive purposes. Although we did not counterbalance question order in any studies, examination of the studies revealed several orders of items. Study 3 included a measure of valence before asking about subjective time. As well, all three precursor studies assessed subjective time at the end of the survey, after a measure of valence (along with other measures) were included. This means that in multiple samples, participants were given a chance to express their attitudes/identity first in a variety of other ways before they were asked about subjective time. In these samples we still find the predicted shifts in subjective temporal distance. We hope that this additional information, in addition to the detailed discussion of this alternative explanation in the discussion, helps allay this concern sufficiently.

We should also note that in your review you suggest that another approach might be to ask about political identity later in the study after key measures. Although we address the response substitution question is several other ways, we don’t directly address this one, for a couple of reasons. First, our studies did not vary order of demographics – partisanship was asked along with a standard set of other demographics (age, gender, religion, household income, etc) and we doubt it would activate partisan identity any more than the vignettes about various political actors and events already would. Second, political identity was asked at the start to avoid another potential problem: asking political identity after the manipulation could lead people to be less inclined to honestly report their politics after reading about their side’s bad behavior. However, we believe that the other responses we provide address the same central problem in slightly different ways consistent with Gal and Rucker’s (2011) recommendations. 

From the discussion (pages 64-65 with tracked changes): 

“With regard to the interpretation of subjective time, it is also important to consider the possibility that the measure was used to express something other than people’s genuine subjective perceptions of time. Researchers have pointed to the methodological challenge of response substitution that can affect survey research in hidden ways [35]. Substitution occurs when participants wish to express attitudes or beliefs that have not yet been asked about and use a seemingly unrelated question to express this view. For example, if a customer is given poor service at a restaurant but the feedback survey only asks about the food, they will give the food a poor rating, regardless of its actual quality, as a means of expressing their negative experience[35]. In the current studies, if participants were asked about subjective time before they have a chance to express their attitudes towards a past scandal, then they could conceivably use the opportunity to report the scandal’s timing as closer or farther as a means of expressing the evaluation they wish to convey. . By this logic, someone might opt to say an event “feels like ancient history” not because it actually seems more distant in time but as a means of expressing “it’s not a big deal.” Indeed, subjective distance measures that incorporate a metaphor like “ancient history” could be especially susceptible to such a concern – people may understand “ancient history” as a dismissal of current relevance, whereas measures simply asking whether an incident seems recent or long ago may be less loaded in this manner. 

Although the concern about response substitution is important, we took several steps to assess this possibility and argue that it is not a likely alternative interpretation of the current findings. First, although in some studies we included the more loaded “ancient history” metaphor as a scale endpoint, it was always asked second after a more subtle measure (“feels very recent/long ago”). Results are largely identical when analyses are limited to the more subtle measure of subjective time (which was always asked first). To offer the most conservative test of our hypotheses, we report results in the manuscript using only the subtle measure of time perception and report results of the two-item scale including “ancient history” in the supporting materials. Second, research on response substitution recommends considering question – specifically, including direct evaluation questions that allow people to express their views first, in order to reduce the likelihood that people will respond expressively on proxy items [35]. 

Although our studies were not designed with this order concern in mind, when reviewing our study methods we note that order of question did vary across studies in ways that help us address the question of response substitution. Specifically, in Studies 1, 2, 4, and 5 the subjective time questions were asked first, but in Study 3 (and all precursor studies reported in supporting materials), subjective time questions were asked after other items that allowed people to express their evaluations and attitudes about the incident. Descriptively, the patterns observed were the same regardless of whether the temporal distance items came immediately or later in the survey. Therefore we argue that subjective distance is more likely to be measuring people’s highly elastic psychological sense of time [4,5] than simply being a proxy for respondents to express their more general approval or disapproval.”

In Study 2, Clinton supporters paradoxically rated both a higher length of the Trump presidency and a longer time until the 2020 election. One interpretation of this is that people associate painful events with the slow subjective passage of time. For a person who finds the Trump presidency highly aversive, much like a root canal, it will feel like it has been going on for a long time and still has a long time to go, or even if it doesn’t actually feel that way, that may be a salient metaphor for expressing their disapproval. I think this could be consistent with either the proposed process or with the attribute substitution alternative. However, I do think these results are inconsistent with a version of the proposed process in which people have a single distorted timeline in their minds (as opposed to the distortion occurring in the context of specific questions).

We agree that Clinton supporters’ longer distance from the Trump presidency and a longer time until the 2020 election are likely due to painful events feeling slower, which is a process that is distinct from the main claim of the paper that people will see an event as closer or farther depending on whether it hurts or helps their political allies or opponents. However, we do not believe it is a contradictory process, as subjective distance and subjective duration are separate phenomena and can each be influenced by separate factors. Recall that subjective distance from the 2016 election was a two-way interaction between voter group and election framing. Clinton supporters saw the negative version of the election as closer than Trump supporters, while Trump supporters saw the positive version of the election as closer than Clinton supporters. By contrast, subjective length of the Trump presidency and distance from the 2020 election only showed main effects of voter identification, with Clinton supporters seeing this political “root canal” as taking longer. Time, indeed, does not fly when you are not having fun, and it’s safe to assume Clinton voters were largely not having fun when reflecting on the Trump presidency, regardless of which framing they read about the election itself. The different patterns for each outcome are not mutually exclusive.

We do agree, however, that the results for subjective length of the Trump presidency and distance from 2020 show a distinct process that would need unpacking if it was included in the main paper, and really is not central to the main theoretical point of the manuscript. Therefore, in an effort to cut down the paper and focus on the most important results, we have removed these analyses from the main paper and moved them to the supporting information. Study 2 now only reports the results for distance from the 2016 election in the main manuscript. 

Was the discussion of manipulating external framing (on p. 50) referring to Study 4? I had a concern about the manipulation in that study. The text says “Mr. Bosch was reported as having an affair outside of his marriage in 2008” and later refers to a 2008 report again. Participants may have interpreted this as meaning that 2008 refers to when the report occurred, not when the affair happened, and then attributed entirely different dates to the affair based on the manipulation. Therefore, the manipulation may represent an actual informational difference, rather than mere framing.

Thank you for seeking clarification of this aspect of the method; it made us realize it should be described more clearly. The only difference between the conditions were the bolded text in brackets shown in the paper (pages 36-37 with tracked changes). The changes included whether Bosch was described as a Democratic or Republican senator, and the phrase at the beginning of the second paragraph which represented the temporal distance manipulation framing [close: Not so long ago, still within this electoral cycle; distant: Quite a number of years ago, some time before being elected Senator; control: No mention]. In all conditions, Bosch’s presidential run (2013) and his affair (2008) were described as occurring at the same time. Even some of the questions refered to “Mr. Bosch's 2008 affair” to clarify the timing refered to the affair, not just the reporting. 

The data did include a memory check that asked when the affair took place. Alhough we did not have any reason to expect it to vary across condition, we checked as a precaution and found that there were no significant differences across conditions or political groups.

I appreciated the open-ness regarding the limitations of Study 5. Given that this was the only pre-registered study, and that multiple aspects of the study did not work as planned to create the conditions in which the hypotheses could be tested (i.e., the focusing manipulation seemingly did not shift responsibility as intended and the verbal fusion scale had a problem), this study did not add as much to the paper. The interaction with identity is consistent with the hypothesis, but since this only occurred with measure not designated as the primary analysis it’s not as conclusive. (As an aside, you may be interested in Chen & Urminsky 2019, which presents a measure of political identity that relates to political attitudes and self-reported behaviors).

We suggest that there is value in a final preregistered study that replicates the key effects in all other studies, and we agree that because of the failed secondary purpose (the focus manipulation) it does not add much new to the package with regard to heightening or reducing the partisan threat – this subtle focus manipulation seems to have been a too-ambitious aim, perhaps especially given the recent and momentous political event we chose to examine. Problems with our identity measures also mean the exploratory analyses are only mildly compelling – and we appreciate that you called our attention to the Chen and Urmisnky measure which we may use as an alternative in future work!

In a general effort to cut down the paper and focus on the most important results, we retain the central preregistered factorial analyses and have removed the exploratory analysis with the group identity measure as a moderator and placed it in the supporting information. The main contribution of Study 5 as it stands it to provide a preregistered replication of our main findings (that people shift time to attack opponents and defend allies) in a novel context (the Capitol riot). The focus manipulation (Republican vs fringe) is of course still included in the main analyses although we did not find any effect.

Overall, I think the current paper makes a strong case that people report events challenging their political views as subjectively more distant. I think the paper makes a weaker case that this represents an actual perception that impacts subsequent attitudes (as opposed to being an outcome of other attitudes or outright attribute substitution). I think this could be addressed by discussing these limitations and toning down some of the claims in the GD. 

Alternatively, I think re-running a pre-registered version of Study 4 with more precise language and some comprehension checks could address these issues. If the latter strategy is taken, I think it could be useful to also include measures of political identity and test the moderation found in Study 5.

We appreciate the overall assessment and added suggestions. As noted in the response to the editor, we addressed response substitution in several ways, and tempered our claims throughout and in the GD to ensure that our conclusions don’t go beyond what we can claim. 

Specifically, throughout the document we take care to be more careful in discussing the causal relationships. All path models have been removed from the main paper and placed in the supporting information (and any claims related to these models are eliminated from conclusions). We have subsequently cut back the talk of “downstream effects” and causal language from the general discussion. Instead, we have added some new statements (pages 60-61 with tracked changes) that makes more tentative claims based on the manipulation in Study 4. Because the manipulation of subjective time revealed a main effect for current relevance, we believe this constitutes some evidence that subjective time can have a causal impact on current relevance. The subjective time manipulation also shifted person perception and hypothetical voting intentions, which we mention as well. We end the paragraph by saying that more experimental work would be needed to fully demonstrate a causal relationship.

We also addressed what we believe to be the central concerns with Study 4, which we were able to address by clarifying the method and reporting a comprehension check on people’s understanding of the timing of the affair. As mentioned in the letter to the editor, we opted not to conduct an additional study (to re-run study 4) because we believe our clarifications alleviate many concerns and because the theoretical contribution of a subjective time manipulation would be modest given Study 4 and past research (e.g., Cortes et al 2018) demonstrating similar effects in interpersonal domains. 

Minor points:

The scenario quoted in Study 3 only shows the negative version.

The positive version of the statements have been added in square brackets (see page 26 with tracked changes).

It would be useful to include the survey instruments in the OSF archive.

Unfortunately because the studies were conducted over a long period of time (including some on a now-defunct survey platform) and before the lab began using OSF regularly, full copies of original surveys are not all available though question wordings, endpoints, and manipulation materials are all available in the open data and supporting materials. If required we can spend some time re-creating old surveys to populate OSF but have not taken that step given that the central information is all available in other places. 

Reviewer 2 Comments

Reviewer #2: This paper reports findings from a series of 5 experimental studies. The studies set out to assess whether motivated reasoning affects subjective assessments of how long ago a politician’s (mostly negative) behavior occurred. The design of the experiments is solid and the replication of core findings across studies is commendable. The core finding—that people tend to view negative behavior from political opponents as more recent, but identical information about an ally as distant—is a useful contribution to our understanding of how people process political information.

The primary limitation of the paper is that far too much extraneous analysis is presented. This leads to bloat that distracts from the core contribution of the paper. The additional analysis is also undertheorized and sometimes feels like it is coming out of nowhere. I would encourage the authors to pare out large swaths of what is presented and either simply exclude this additional material or confine it to an appendix. Consider Study 5. This is the fifth study on the theme presented in the paper. My read is that one contribution is to simply use a very high salience, recent event (January 6 insurrection). The other is that the experiment varies whether the events were framed as being precipitated by “fringe groups” or the Republican Party more generally. If so, it seems like this section should be concise, noting similarities in findings to Studies 1-4 and describing what the “fringe/mainstream” treatment adds. Instead the section of the paper on this study stretches from page 37 to page 49.

Much of this bloat is driven by exploration of how the treatments affected an array of outcome variables beyond subjective perceptions of how long ago an event happened. At various points, stray potential moderators like media consumption are brought in. Table 7 alone takes up almost a page exploring how the treatments affected (or didn’t affect) perceptions regarding who was responsible for the riots. Table 8 pursues this further reporting marginal means by condition for each of 9 potential entities who respondents might view as responsible. In a sense this seems like a manipulation check (did focusing on extremist groups lead, say, Republican respondents to view the events as less about “their group’s” behavior?) If so, this analysis should not be allotted multiple pages of the paper, while discussion of the core question gets half a page (p. 45). Skip to the analysis that matters and note that the null effect of the focus treatment here may be a product of that treatment failing to meaningfully shift how people thought about who “did it.” Refer readers to an Appendix for more detail.

We appreciate the forthright (and fair!) assessment that the paper is far too long for its contribution value. We agree that the paper can make a clearer (and more readable) contribution if we maintain forcus on the central questions. Most of the extraneous analyses have been removed and placed in the supporting information. Removed sections include the following.

• All path analyses

• The “Other Measures of Subjective Temporal Distance” and “Other Mechanisms of Subjective Temporal Distance From Election” in Study 2 

• Results for the person perception and voting intentions variables in Studies 3 and 4 are reported briefly and largely confined to tables. 

• The “Attitudes Regarding the Capitol Storming”, “Assigned Responsibility”, and “Group Identity” sections have been removed from Study 5 

This means that across all five studies, the results are mainly limited to manipulation checks, subjective time, and current relevance. 

I am sympathetic to the authors’ impulse to try to use all of the data and fit a bunch of pieces together. My read is that these efforts raise far more questions than they answer. For example, I understand that the authors would like to show that subjective temporal distance perceptions affect vote choices. However, these data do not allow them to demonstrate that. Consider the path analysis reported in Figure 4. This model rests on critical *assumptions* about the chain of causality. I am skeptical that the model proposed is actually how things work. Why should we believe that group attachment to a political figure affects subjective temporal distance, which in turn affects perceived relevance, which in turn affects assessments about the political figure’s morality (etc), which in turn affects vote intentions? It seems at least as plausible to posit that people like candidates/figures from their preferred parties and are inclined to vote for them. When confronted by bad behavior from the past, they may rationalize their vote intentions by reasoning that the event isn’t really such a big deal and, after all, it happened quite a while ago, didn’t it? I would suggest cutting the path analysis from the paper as it implies that the correlations between post-treatment variables can be used to shed light on the chain of causality. They cannot.

We agree that the path models cannot establish a chain of causality and that there is not a particularly strong rationale for the order of variables we specified. The path models have been removed from the paper (and placed in the supporting information with appropriate caveats and without claims about causality). As well, we have cut back the talk of “downstream effects” and causal language from the general discussion (see pages 60-61 with tracked changes).

We do, however, make some tentative claims about the causal influence of subjective time on the basis of the subjective time framing manipulation in Study 4. In a new paragraph in the general discussion (pages 60-61 with tracked changes) we note that the manipulation of subjective distance did have an effect on current relevance, as well as the person perception and voting intention variables. We dropped the path models in Study 4, which both took away from and likely obscured these basic experimental effects testing the causal impact of subjective time. 

In short, this could be a strong, focused 20-25 page paper that offers new insights into how motivated reasoning affects how people respond to information about instances of past wrongdoing. Existing work (e.g., Pereira and Waterbury 2019; Doherty, Dowling, Miller 2014) shows that the effects of scandals fade with the passage of time. This paper offers new insights into how partisans may reason about past events like this. If the authors want to show that subjective assessments of temporal distance are correlated with other judgments (e.g., vote intent; perceived relevance to current evaluations), they certainly can do so. However, they should be agnostic about the chain of causality.

As noted previously we have shortened the paper and have taken care to avoid unwarranted claims about causal chains. Also, thank you for pointing to these papers on political scandals! These political science papers were not on our radar and are relevant to the current research (though they approach these questions in very different ways). We have added a paragraph to the introduction (pages 3-4 with tracked changes) discussing this past research and how subjective time might shed light on the outstanding questions, returning briefly to it in the discussion (page 59 with changes) to highlight that our work may shed some light on how people process such political scandals.

*** Additional notes

The revisions I suggest above mostly involve cuts. I would suggest the authors use some of this “room” to provide a bit more detail about the treatments used in Studies 1-3. Where did the text of these treatments come from? Should we be concerned that the text of the treatments is so different across conditions?

Thanks for flagging that these treatments could benefit from greater clarity. The text of the treatment conditions in Study 1 is included in Table S5 (supporting materials). To the main paper I added “All incidents were real, but the descriptions were written for the study (see Table S5 for the text contained in both conditions)” in order to clarify that the passages were not taken from any other specific source (page 9).

The text in Study 2 is included in Table S7. I added “passage created for the study” in the main paper to clarify that it wasn’t taken directly from somewhere else (page 15).

In Study 3 we already provide the exact text, state that it was fictional, and show that very little changed between conditions (pages 26-27).

We appreciate the question about differences between the text of treatment conditions. Individual studies may be critiqued for lack of perfect parallel between conditions. This reflects the trade-off between the ecological validity of using real-world events and actors, and the additional experimental control gained using artificial stimuli that allow much closer parallels. We suggest that the diversity of methods across studies helps to allay concerns about individual studies because they all have different strengths and limitations yet converge on the same pattern. 

Similarly, I’d like to see a bit more about sample restrictions. E.g., on page 8 the authors refer to participants “whose bias could not be reasonably guessed.” How did the authors try to “guess”? 

The original passage was worded in a misleading manner. We did not try to guess the loyalty of the independent and undisclosed participants, because we did not think that they could reasonably be discerned. The passage that implies we did try to guess has been removed.

And exclusions tied to attention checks should consistently explain what the attention check was. In some cases the exclusions are substantial (e.g., 623 get pared to 327 in Study 2). The reader needs to know how many lost cases each exclusion results in.

Thanks for noting the inconsistent reporting. Attention checks were used in Studies 1, 2, and 5, and asked participants to report specific number or corresponding response option (e.g., “To show you are paying attention to these questions, please select “strongly agree” for this question.”). The “Participants and Power” section in each of these studies now specifies that these were attention checks “instructing participants to select a specific number.”

We have also now reported how many exclusions were due to various factors. Because exclusion criteria were not mutually exclusive (some people might be coded as “exclude” based on more than one exclusion criteria), the numbers of reported exclusions do not always add up to exactly the same as the number of participants dropped from analyses. 

We acknowledge that some studies have a fairly high exclusion rate. There are a few reasons for this. First, some studies were conducted on MTurk during periods where data quality was quite low and bots were expected, and rather than screening for quality in advance (as later developments in participant platforms have allowed us to do) we simply accepted high exclusions. Second, we intentionally recruited people without screening for political party in advance (we did not want to advertise that only Democrats and Republicans were being recruited, for example, because this could draw too much attention to partisan identity and create suspicion about the purpose of the study. Therefore we knew a prior that if our studies required categorization as political partisans, supporters or opponents, some participants would need to be excluded because they identified otherwise. 

Having said that, we also appreciate that you called attention to Study 2 which does seem to have an unusually high exclusion rate. We investigated and realized that we had made an error in reporting; the original data set included additional participants who weren’t assigned to the three experimental conditions and incorrectly inflated the apparent sample size. The proper initial sample size was N = 458, thus the exclusion rate, though still non-neglible, was not so dramatic. The error occurred because the study passed from one lab alumna and co-author (Williams) who collected the data to another (Dawson) who analysed and reported it. In fact, a mini-pilot study had been conducted for another purpose during the same wave of data collection, and for convenience was included as a “fourth” condition, but was never intended to be part of the current study. To be fully transparent, the fourth condition was similar to the control but asked questions in a different order (starting with legitimacy judgments) as a preliminary check to see if counterbalancing would matter in future studies. This counterbalance was not incorporated into the full study design and was intended a priori to be excluded from Study 2. We apologize for the oversight and are grateful for the review noticing this anomaly. We would also like to note a correction we made when re-examining all of the data sets to report the exclusion criteria in more detail. In Study 4, we originally report the initial sample was 420 when it should have been 407. This error is corrected in the text and has no implications for final sample size or analyses.

Tables tend to be easier to interpret if their title and/or notes are clear about what variable is being analyzed (e.g., Table 1)

Apologies for the ambiguity. The current tables should now all clearly state which variables are reported.

Are the power tests reported the right ones? They seem to pertain to direct effects, but the estimates of interest are almost all (all?) tied to interactions.

All sensitivity analyses refer to the predicted interactions, and we checked the text and clarified where relevant. In Study 1 in particular we have changed the wording to specify that the sample was sufficient to “detect an interaction with an effect size of…”.

Cut the snarky first sentence from the General Discussion.

Good call. Noted, and removed. Sometimes what seems like light humor in a moment of flourish can easily come across as snark, and in any case likely would not stand the test of time! 

Reviewer 3 Comments

Reviewer #3: This paper investigates a phenomenon that can be related to the recent literature on motivated beliefs. People perceive bad past events more in the past than good events. Specifically, they think that bad behaviors of supported politicians are more in the past than bad behaviors of not supported politicians. The study uses well-powered surveys conducted with MTurk. The dependent variable is based on two qualitative questions on how much in the past people think certain events are. The main effect (an interaction of whether one favors the politician and whether the event is positive or not) was found in five independent studies. The individual studies vary in whether they study real or fictitious politicians, in the events and some of the studies introduced a manipulation.

The research question is interesting and given that the effect could be shown in five independent experiments it appears as robust. The phenomenon is relevant because it identifies a mechanism for a bias in favor of preferred candidates. However, it is not clear how a debiasing strategy could look like. The analysis is comprehensive and makes clear what is in the focus and what has to be considered as more explorative.

Thank you for the positive feedback! Debiasing techniques would be an interesting question for future research, though beyond the scope of the current research. 

Specific comments

1. The structure follows the standard of psychology papers presenting the studies sequentially. This structure is less common in other fields such as economics. Since the studies differ in the variables that are collected, this is not a bad structure. Nevertheless, I recommend adding more integrating information. In particular, a summary table reporting all studies would be helpful.

We appreciate this suggestion of summary tables, and suspect it was motivated in part by the manuscript length and the fact that key findings were sometimes obscured by extraneous detail. Consistent with reviewer and editor feedback we chose to shorten and focus the manuscript, which may reduce the need for additional summary tables. In the interest of minimizing manuscript length, we opted not to include additional summary tables, though we can do so should it be requested. 

2. Tables and figures are not self-contained. I recommend adding more information. Further, even if this is irrelevant for the published paper, it would be helpful for the referees if there were no page breaks within tables.

Noted, apologies for the ambiguity. The current version of each table should clearly specify the contents and variables involved in the titles and headings. Notes have been added to the means tables to specify that numbers in parentheses correspond to standard errors.

3. Study 2. In order to classify people as Clinton or Trump supporter, people were asked to report whom they voted for, and those who were “very dissatisfied” with their vote at present were excluded. I wonder why they did not directly ask for the current support for these candidates. Excluding always reduces the number of observations and, if not pre-specified, adds arbitrariness in the analysis.

Thanks for this feedback. As in all studies, political allegiance was the key categorization goal. Across studies, we always recognized a priori that people who could not be adequately categorized into partisan groups (or supporter/opponent) would have to be excluded (see response to Reviewer 2 on pages 15-16 of this letter for additional discussion of exclusions). These studies were run prior to the norms we now have adopted for planning exclusions a priori, and usually preregistering them. The current package of studies, run by multiple students over more than a decade, did not follow these best practices we now know to value! 

Appropriate exclusions to capture the a priori intended variable of partisan allegiance was tricky in Study 2 because people reflected on the 2016 election on its one year anniversary. We asked the standard questions about voting (allowing us to classify partisanship at the time of the election, certainly one relevant factor). In retrospect, we realize that current support for the candidates would also have been a good variable to include in the survey (though even with this variable, both their vote at the time and their current feelings would still be relevant to categorizing allegiance). However, since current support for candidates was not included, we thought that current dissatisfaction with their earlier vote was not a bad proxy to capture current party allegiance (a Trump voter who now regrets that vote is unlikely to show partisan allegiance to Trump, and vice versa for Clinton). Although we argue that the exclusion is theoretically reasonable we also acknowledge the inherent “researcher degrees of freedom” problem. It is worth noting that if those participants who were excluded for being dissatisfied with their vote were included in the data set on the basis of their original partisanship, the key results do not change meaningfully. 

4. Page 19, “we did not necessarily predict the same interaction”. This formulation feels strange. I believe the authors that their focus variable was the subjective distance from the 2016 election, and that the other variables were more explorative. Nevertheless, I would explicitly consider this as explorative analyses.

5. Page 21, analysis of the post-election media exposure × voter affiliation regression. The report of the regression coefficients in the text are problematic, because the main effects depend on the how the other variable is defined (omitted category). I would prefer to see the regression information in a table and that the discussion of the regression more quickly starts with the slope analysis.

In responding to other reviewer comments about the length of the paper and number of exploratory and extraneous analyses, we cut out most of the analyses that did not pertain to the manipulation checks, subjective time, and current relevance and moved them to the supporting information. The comments above refer to analyses that have been removed from the paper.

6. Obvious malingerers were excluded in study 3. Were similar exclusions possible in the other studies.

In Study 3 we neglected to include attention checks, but noted in examination of the data a small number of participants (n=8) showing clear patterns of inattentive responding or missing data on key DVs. Thank you for calling our attention to the term we used (malingerers) which is really not the clearest way to describe the criteria (we now simply say inattentive responding and missing key DVs). 

Minor issues

7. In Table 1 valence is negatively coded, i.e. bad is positive. This is confusing, in particular because it seems different in Table 4.

The valence measure has been reverse-coded so that bad is now negative, just like in Study 3.

8. I recommend putting the middle category into the middle in tables. For example, in table 5, the comparisons would be more natural if the order would be Close – Control – Distant and Supporter – Neutral – Opponent.

We appreciate the suggestion. In Tables 2, 4, 6 the middle category is now in the middle of the table.

9. In the beginning of the general discussion, some statements were stroked though. What is the status of these statements? Anyhow, the beginning of this chapter is not so convincing.

The strike-throughs at the start of the general discussion were a poor attempt at humor; the feedback is appreciated! The statements and the opening sentence have been removed and the introduction to the general discussion revised.

Typo

10. Page 6: Replace “In an effort to be maximally transparent, we include three studies, from a prior paper investigating implicit theories of change [14], are included in the supporting information to this paper.” by “In an effort to be maximally transparent, three studies from a prior paper investigating implicit theories of change [14] are included in the supporting information to this paper.”

This has been changed.

Page 25, “supporting informaiton”

This statement was removed when we cut out the relevant analyses.

Once again we thank the editor and reviewers for their helpful feedback. We hope that our response reflects how seriously we took these considerations and our efforts to address them. We believe that the review process has substantially improved the manuscript and we are grateful for the opportunity to have this revised contribution considered for publication in PLOS ONE. 

Best regards,

Andrew Dawson (on behalf of the research team)

---

## [Decision Letter · Decision Letter 1]

19 Sep 2022

PONE-D-22-13208R1Far away or yesterday? Shifting perceptions of time for political endsPLOS ONE

Dear Dr. Dawson,

Thank you for submitting your manuscript to PLOS ONE. After careful consideration, we feel that it has merit but does not fully meet PLOS ONE’s publication criteria as it currently stands. Therefore, we invite you to submit a revised version of the manuscript that addresses the points raised during the review process.

The three reviewers from the previous round read the paper again. Two of the reviewers recommended acceptance and one recommended a minor revision. I agree that there is still a small amount of work left, which is why I am considering this a minor revision. Please note that this means that I will NOT send the paper back out to review, but will rather make the final determination on my own. I encourage you to read over the reviews (especially R1) and address the final concerns that were raised. I don't anticipate that this will be a difficult hurdle for you. Congratulations on an important contribution to science.

We look forward to receiving your revised manuscript.

Kind regards,

Jeff Galak, PhD

Academic Editor

PLOS ONE

Journal Requirements:

Reviewers' comments:

Reviewer's Responses to Questions

**Comments to the Author**

1. If the authors have adequately addressed your comments raised in a previous round of review and you feel that this manuscript is now acceptable for publication, you may indicate that here to bypass the “Comments to the Author” section, enter your conflict of interest statement in the “Confidential to Editor” section, and submit your "Accept" recommendation.

Reviewer #1: (No Response)

Reviewer #2: All comments have been addressed

Reviewer #3: All comments have been addressed

2. Is the manuscript technically sound, and do the data support the conclusions?

Reviewer #1: Yes

Reviewer #2: Yes

Reviewer #3: Yes

3. Has the statistical analysis been performed appropriately and rigorously? 

Reviewer #1: Yes

Reviewer #2: Yes

Reviewer #3: Yes

4. Have the authors made all data underlying the findings in their manuscript fully available?

Reviewer #1: Yes

Reviewer #2: Yes

Reviewer #3: Yes

5. Is the manuscript presented in an intelligible fashion and written in standard English?

Reviewer #1: Yes

Reviewer #2: Yes

Reviewer #3: Yes

6. Review Comments to the Author

Reviewer #1: I appreciated the revision, and the discussion of response substitution. I think the paper is clearer and reads better now. However, I have a few remaining concerns.

I agree that subjective distance and subjective duration can be psychologically distinct, and that may explain the seemingly discrepant results in Study 2. That said, I don’t think removing the findings about subjective duration is an appropriate way to deal with the issue. I see it as a potentially failed prediction of the theory that can also be explained (post-hoc) by another mechanism, and I think it should be discussed as such, so that the reader can form their own interpretation.

Beyond the issue of response substitution, which I think has been reasonably addressed in this revision, there is a question of the interpretation of the effect. Specifically, it is unclear whether the observed findings are due to explicit motivated reasoning (e.g, temporal distancing as a tactic to minimize the impact of uncomfortable facts) or an implicit process outside of conscious awareness, or perhaps some form of dissonance reduction that might lie in between. I think the paper, as currently written, heavily implies an implicit process, without providing the evidence needed to draw that conclusion. First, I think it would be helpful to go through the paper and carefully consider the wording – the evidence only measures what people choose to report, not what they actually think or feel. Second, I think the speculation in the general discussion (“an innocent, subtle and possibly automatic process”, p. 35; “subtly shaped” and “potential subtle impact”, p. 36) is one-sided. I think it should be either be omitted or balanced out with a discussion of the opposite possibility (“a motivated, direct and possibly deliberate distortion”).

I understand that Study 5 was pre-registered, but I was not able to find the pre-registration (I may have missed it). I didn't see a link in Study 5 or a posted pre-registration on OSF.

Lastly, the OSF files include SAV and SPS files. I would recommend uploading .csv data files and a brief data dictionary explaining what each variable is and what the response codes are. I understand some of the survey materials are no longer available, but I think whatever materials can be found should be uploaded. Ideally, the OSF should contain all the information needed to either re-analyze the data or re-run the studies.

Reviewer #2: (No Response)

Reviewer #3: The revised paper addressed almost all the point I raised. The only point that was ignored was that no table summarizing all the treatment was included. In the end, it is a matter of taste whether such a table is considered as useful, and the argument against the table is OK. The paper has gained in focus and the significant shortening was essential to improve the paper. I spotted a typo on line 791 (two periods) but apart from this I consider the paper as publishable.

7. PLOS authors have the option to publish the peer review history of their article (what does this mean?). If published, this will include your full peer review and any attached files.

Reviewer #1: No

Reviewer #2: No

Reviewer #3: No

---

## [Author Response · Author response to Decision Letter 1]

20 Oct 2022

Our full response can be found in the included "Response to Reviewers October 2022" document. We have copied the contents below.

---

## [Editor Report · Decision Letter 2]

24 Oct 2022

Far away or yesterday? Shifting perceptions of time for political ends

PONE-D-22-13208R2

Dear Dr. Dawson,

We’re pleased to inform you that your manuscript has been judged scientifically suitable for publication and will be formally accepted for publication once it meets all outstanding technical requirements.

Kind regards,

Jeff Galak, PhD

Academic Editor

PLOS ONE